# IL-7 receptor blockade blunts antigen-specific memory T cell responses and chronic inflammation in primates

Lyssia Belarif[1,2], Caroline Mary[1,2], Lola Jacquemont[1], Hoa Le Mai[1], Richard Danger[1], Jeremy Hervouet[1], David Minault[1], Virginie Thepenier[1,2], Veronique Nerrière-Daguin[1], Elisabeth Nguyen[1], Sabrina Pengam[1,2], Eric Largy[3,4], Arnaud Delobel[3], Bernard Martinet[1], Stéphanie Le Bas-Bernardet[1,5], Sophie Brouard[1,5], Jean-Paul Soulillou[1], Nicolas Degauque [1,5], Gilles Blancho[1,5], Bernard Vanhove[1,2] & Nicolas Poirier[1,2]

Targeting the expansion of pathogenic memory immune cells is a promising therapeutic strategy to prevent chronic autoimmune attacks. Here we investigate the therapeutic efficacy and mechanism of new anti-human IL-7Rα monoclonal antibodies (mAb) in non-human primates and show that, depending on the target epitope, a single injection of antagonistic anti-IL-7Rα mAbs induces a long-term control of skin inflammation despite repeated antigen challenges in presensitized monkeys. No modification in T cell numbers, phenotype, function or metabolism is observed in the peripheral blood or in response to polyclonal stimulation ex vivo. However, long-term in vivo hyporesponsiveness is associated with a significant decrease in the frequency of antigen-specific T cells producing IFN-γ upon antigen restimulation ex vivo. These findings indicate that chronic antigen-specific memory T cell responses can be controlled by anti-IL-7Rα mAbs, promoting and maintaining remission in T-cell mediated chronic inflammatory diseases.

[1] Centre de Recherche en Transplantation et Immunologie (CRTI) UMR1064, INSERM, Université de Nantes, Nantes 44093, France. [2] OSE Immunotherapeutics, Nantes 44200, France. [3] Quality Assistance, Thuin 6536, Belgium. [4] ARNA laboratory, Université de Bordeaux, INSERM U1212, CNRS UMR5320, IECB, Bordeaux 33076, France. [5] Institut de Transplantation Urologie Néphrologie (ITUN), CHU Nantes, Nantes 44093, France. These authors contributed equally: Lyssia Belarif, Caroline Mary. These authors jointly supervised this work: Gilles Blancho, Bernard Vanhove, Nicolas Poirier. Correspondence and requests for materials should be addressed to N.P. (email: nicolas.poirier@ose-immuno.com)

Therapeutic targeting of proinflammatory cytokines has demonstrated clinical benefit in several immune-mediated disorders. However, drugs that target downstream mechanisms of dysregulated immune responses (e.g., TNF), are not effective in all patients or diseases, depend on specific etiologies, and significant rates of primary and secondary resistance are still observed. Novel therapeutic approaches targeting more upstream mechanisms are desired to prevent relapse and maintain long-term remission. Several genome-wide association studies have identified IL-7R alpha chain (IL-7Rα) polymorphism as one of the first non–major histocompatibility complex–linked risk loci for susceptibility of multiple sclerosis[1–3], type 1 diabetes[4,5], inflammatory bowel diseases[6], rheumatoid arthritis[7], systemic lupus erythematosus[8], atopic dermatitis[9], and sarcoidosis[10].

Interleukin-7 (IL-7) is a limiting and non-redundant cytokine that is mainly produced by epithelial and stromal cells and regulates T cell homeostasis, proliferation, and survival[11,12]. Conventional mature T lymphocytes express high levels of the IL-7 receptor (IL-7R), with the exception of naturally-occurring regulatory T-cells (Tregs) that express low IL-7R. This constitutes a unique opportunity to selectively target pathogenic effectors while preserving natural regulators[13–15]. IL-7 signals through the cell-surface IL-7R, formed by the dimerization of the IL-7Rα (CD127) and the common cytokine receptor gamma chain (γ-chain, CD132)[16]. As depicted in Fig. 1, IL-7 interacts with both domain D1 of the IL-7Rα (site-1) and domain D1 of the γ-chain subunit (site-2a); IL-7Rα and the γ-chain also interact together with their D2 domains (site-2b), stabilizing and forming an active IL-7/IL-7Rα/γ-chain ternary complex[17–19]. IL-7R activation induces proliferative and anti-apoptotic signals mainly by activating the JAK-STAT pathway. Some studies have reported that IL-7 can also activate the PI3K or MAPK/ERK pathways, suggesting that IL-7 could use different signaling pathways depending both on cellular type and the physiological status of the cell[11,20].

Transgenic mice overexpressing IL-7 spontaneously develop chronic colitis with infiltration of IL-7Rα^high colitogenic T cells in the gut mucosa[21–23] and also develop a chronic dermatitis infiltrated mainly by γδ T cells[24,25]. Similarly, IL-7 produced by hair follicular keratinocytes is required for skin-resident memory T cells, epidermotropism and contact hypersensitivity responses in mice[26]. Blockade of IL-7Rα consistently demonstrated potent efficacy in preventing and/or curing ongoing diseases in several autoimmune and chronic inflammatory rodent models, including type 1 diabetes[27,28], multiple sclerosis[29,30], rheumatoid arthritis[31–33], colitis[22,34], systemic lupus erythematosus[35], and primary Sjögren's syndrome[36]. We and others also demonstrated the efficacy of anti-IL-7Rα mAbs in inducing long-term survival of islet or skin allografts in rodent transplantation models[37,38].

Altogether, studies in rodent models depicts IL-7 as a fuel driving chronic autoimmune and inflammatory diseases[39]. However, in accordance with the non-redundant role of IL-7 for T-cell homeostasis and the severe immunodeficient phenotype observed in IL-7 or IL-7Rα knock-out mice[40,41], the efficacy of anti-IL-7Rα mAbs in rodents is generally associated with the induction of lymphopenia impacting both naive and memory T and B lymphocytes. No translational studies in higher species have yet confirmed the therapeutic potential of targeting IL-7Rα. Only one recent study in the common marmoset monkey has reported that a blocking anti-IL-7Rα mAb was efficient ex vivo in inhibiting IL-7-induced STAT5 phosphorylation. However, it did not induce any modification in peripheral lymphocyte counts and was not able to significantly prevent the clinical disease course of neuropathology in an experimental autoimmune encephalomyelitis (EAE) marmoset model[42].

In this study, we report that therapeutic efficacy of blocking anti-human IL-7Rα mAbs in a non-human primate memory T-cell-induced chronic inflammation model depends on mAbs epitope property, in particular at the receptor heterodimerization region (site-2b) and on the capacity to thoroughly inhibit IL-7R signaling. We find that all tested IL-7 blocking anti-IL-7Rα mAbs prevent JAK/STAT signaling but, depending on the epitope, some mAbs also present agonist properties that modify the transcriptome of peripheral blood mononuclear cells (PBMCs). These IL-7 blocking mAbs with dual agonist and antagonist properties lack efficacy in vivo. By contrast, an 'antagonist-only' anti-IL-7Rα mAb induces long-term control of skin inflammation in non-human primates, by neutralizing or depleting IFN-γ producing antigen-specific memory T cells, but without inducing lymphopenia or polyclonal T-cell functional or metabolic defects as generally observed previously in rodents[37].

## Results

**Anti-IL-7Rα mAbs control chronic skin inflammation**. We generated and screened new anti-human IL-7Rα mAbs for their ability to antagonize IL-7-induced STAT5 phosphorylation on human peripheral T lymphocytes in vitro. Epitope characterization by linear peptide array of lead mAbs identified two types of anti-human IL-7Rα antagonist mAbs: (1) mAbs binding to the region (site-1) of interaction with IL-7 (Fig. 1), as previously reported by other groups (patents WO/2011/094259 and WO/2011/104687), and (2) a mAb which binds both site-1 and an epitope overlapping to the domain (site-2b) of heterodimerization between IL-7Rα and the γ-chain subunits[19]. Two mAbs were recombinantly generated from the same site-1/2b binder, with a human IgG4 Fc isotype (containing the S228P hinge mutation to prevent Fab-arm exchange as previously described[43,44]) or a human IgG1 isotype to evaluate its potential effect on cytotoxicity. Another mAb was recombinantly generated from a site-1 binder with the same IgG4 isotype. These three recombinant anti-IL-7Rα mAbs exhibited similar affinities (KD of $2 \times 10^{-10}$ M for the binder to site-1/2b in IgG4 isotype, $3 \times 10^{-10}$ M for the site-1/2b mAb in IgG1 format and $5 \times 10^{-10}$ M for the binder to site-1 in IgG4 format) and importantly displayed highly similar dose-response inhibition of IL-7-induced STAT5 phosphorylation on baboon T lymphocytes in vitro (Supplementary Figure 1A). IL-7Rα interacts also with the TSLPR chain to form the heterodimeric receptor of the TSLP cytokine. None of these three mAbs significantly prevented TSLP receptor signaling as measured by less than 20% of inhibition of TSLP-induced CCL17 secretion by dendritic cells (Supplementary Figure 1B). As expected, only the IgG1 mAb induced significant antibody-dependent cellular cytotoxicity (ADCC) on IL-7Rα+ target cells incubated with NK cells in vitro (Supplementary Figure 1C).

These three anti-IL-7Rα mAbs were administrated intravenously (10 mg/kg, $n = 3$ per antibody) in baboons pre-sensitized with bacillus Calmette–Guérin (BCG) vaccine to generate an antigen-specific memory T lymphocyte response. Intradermal injection of tuberculin-purified protein derivative (PPD) after vaccination elicited a robust and highly reproducible delayed-type hypersensitivity (DTH) reaction characterized by a memory Th1-mediated skin inflammation as previously described[45,46]. All animals presented a measurable erythema response (maximum erythema response > 10 mm) and high infiltration of T lymphocytes and macrophages in the challenged skin biopsies after a first tuberculin intradermal challenge (IDR) (Fig. 2a, b). After a washing-out period of 4 weeks, a second tuberculin IDR was performed 24 h after anti-IL-7Rα mAb administration. Administration of the site-1/2b anti-IL-7Rα mAbs dramatically prevented memory T cell-mediated skin inflammation after

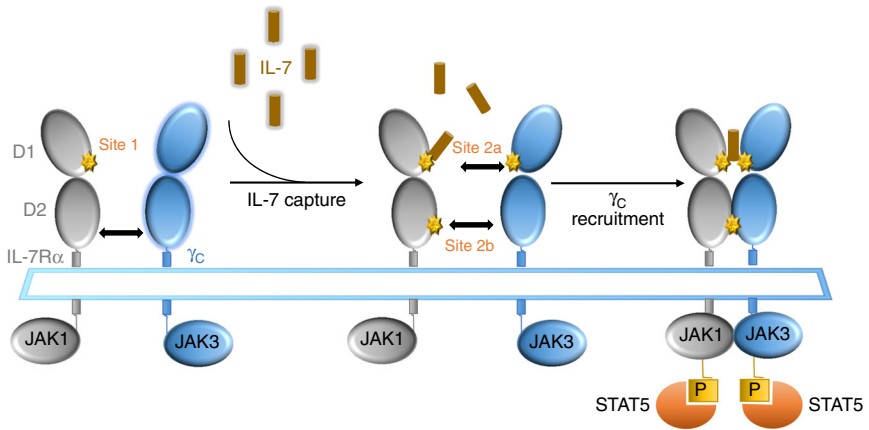

**Fig. 1** Schematic representation of cytokine-induced receptor heterodimerization signaling mechanisms as previously proposed[19]. During the initiation step, IL-7 interacts with the extracellular domain 1 (D1) of IL-7Rα, generating the *site-1* interface. This leads to the intermediate step where a 1:1 complex can associate with the shared common gamma-chain (γc) receptor. The binding of γc receptor involves an interface between IL-7 and γc called *site-2a* and an interface between D2 regions of the IL-7Rα and γc receptor called *site-2b*. The stabilized heterodimer complex activates the JAK/STAT and possible other signaling pathways

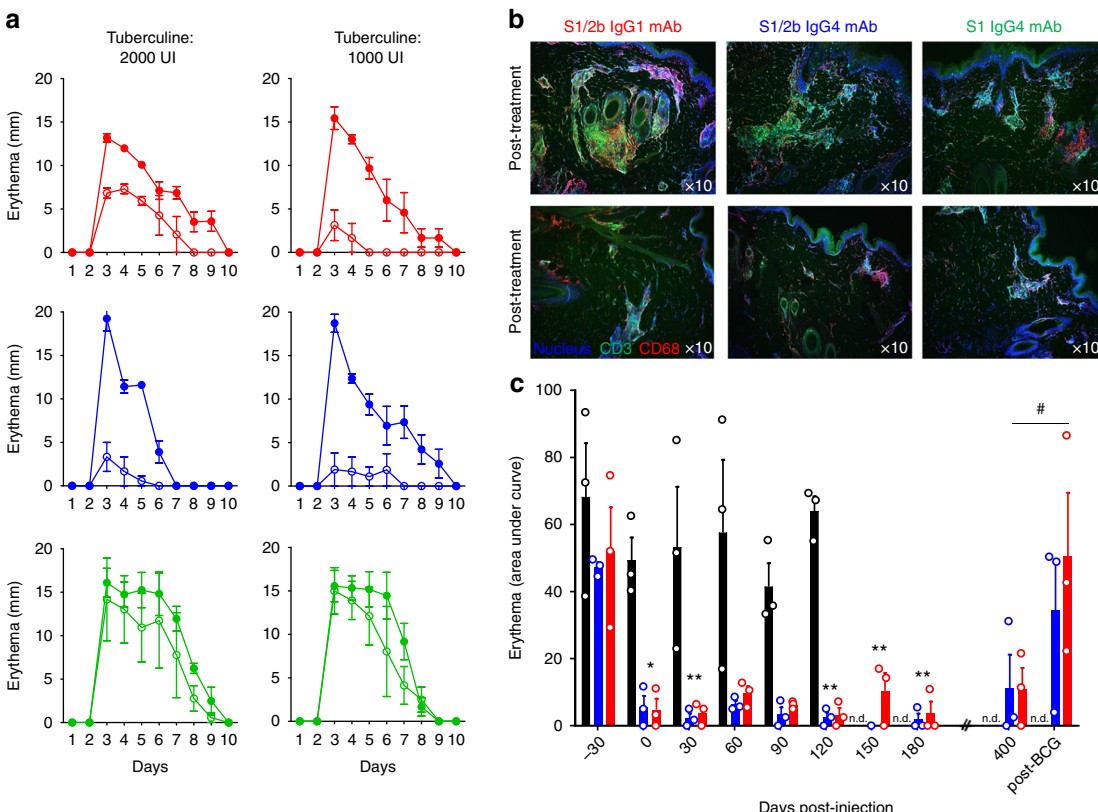

**Fig. 2** Anti-IL-7Rα mAbs induce long-term control of skin inflammation. **a** Cutaneous erythema diameters measured daily after tuberculin intradermal injection in baboons treated with a single intravenous injection with 10 mg/kg of the site-1/2b IgG1 mAb (red, *n* = 3), site-1/2b IgG4 mAb (blue, *n* = 3) or site-1 IgG4 mAb (green, *n* = 3). Control erythema (closed symbols) was performed 4 weeks before treatment. Baboons were then rechallenged 24 h after mAb treatment (open symbols). **b** Representative skin immunostaining (green: CD3, red: CD68, blue: nucleus) 4 days after tuberculin challenge performed a month before or 24 h after mAb injection as described in (**a**). **c** Area under curve (AUC) of cutaneous erythema diameters measured after tuberculin intradermal challenges performed monthly or at the indicated time in baboons treated with site-1/2b mAb (IgG1: red and *n* = 3, IgG4: blue and *n* = 3) as in (**a**), or in historical placebo-treated control animals (black histograms) following the same protocol[45]. n.d. not determined in control animals. Data are mean ± SEM and erythema diameters were measured by at least two observers. *$p < 0.05$; **$p < 0.01$ one-way ANOVA within site1/2b treated animal group across all timepoints as compared to pre-treatment responses. # $p < 0.05$ Mann–Whitney between pre-re-vaccination and post-re-vaccination

antigen rechallenge as displayed by a marked reduced erythema response (Fig. 2a–c) and immune cell skin infiltration (Fig. 2b) in all baboons ($n = 3$ with IgG1 mAb and $n = 3$ with IgG4 mAb). The site-1/2b mAb in its IgG4 format was even more efficient compared to the IgG1 isotype with a higher dose of tuberculin administration. In contrast, none of the three baboons treated with the site-1 anti-IL-7Rα mAb displayed a reduced erythema response, in spite of a similar pharmacokinetic profile of the antibody and a similar occupancy of the receptors (Supplementary Figure 2B–C). Similarly, primary immunization against sheep red blood cells (SRBC) was significantly reduced in the six animals treated with site-1/2b anti-IL-7Rα mAbs, while baboons treated with the binder to site-1 displayed no difference as compared to control animals (Supplementary Figure 3A).

Complete blood cell counts and immunophenotyping analyses were performed during the period of exposure to the mAbs. Administration of anti-IL-7Rα mAbs binding to site-1 or site-1/2b did not significantly change lymphocyte, monocyte or granulocyte blood cell counts even with the IgG1 mAb which otherwise demonstrated ADCC properties in vitro (Supplementary Figure 4). Similarly, the frequency of T and B cells in the peripheral blood remained unchanged during the 1-month period of evaluation and no difference was observed between the three anti-IL-7Rα mAbs. Further analysis of T cell subsets did not reveal any significant modification of naive, effector or central memory subsets within CD4$^+$ or CD8$^+$ T cell populations. No significant increase in natural CD4$^+$ regulatory T cells (Tregs) or activated/exhausted CD8$^+$ T cells (based on PD-1 expression) was observed (Supplementary Figure 4). Finally, none of these anti-IL-7Rα mAbs elicited significant cytokine release following drug administration (Supplementary Figure 5). Only a modest IL-6 release (<200 pg/ml) was detected with the three mAbs within 4 to 8 h after injection, coinciding however with the per protocol immunization with SRBC i.v. The only immunophenotyping difference observed between site-1 and site-1/2b anti-IL-7Rα mAbs was a transient decrease in IL-7Rα expression on the surface of blood T cells with the binder to site-1 (Supplementary Figure 2A) as previously described by others with another anti-IL-7Rα mAb[47].

Placebo-treated control baboons developed consistent repetitive erythema responses after monthly tuberculin challenges during the five-month period of evaluation (Fig. 2c). Responses to tuberculin challenges remained significantly inhibited (both in terms of intensity and duration of erythema) for a very long-term period (>1 year) in the 6 baboons treated only once before the second antigen challenge with site-1/2b anti-IL-7Rα mAbs (Fig. 2c), and lasted for several months after complete drug elimination (up to 40 terminal half-life) as analyzed in the serum and on the surface of blood or lymph node T cells (Supplementary Figure 2 B–C). These animals, however, remained immunocompetent as demonstrated by their ability to mount an IgG response upon re-challenge with SRBC 5 months after drug administration (Supplementary Figure 3B). Furthermore, a second cycle of vaccination with BCG vaccine restored an erythema response to intradermal tuberculin challenges in these animals confirming their immunocompetent status and suggesting that the long-term antigen-specific immune tolerance observed here was reversible (Fig. 2c).

**Antagonist and agonist properties of anti-human IL-7Rα mAbs**. The unexpected different in vivo therapeutic effect of anti-human IL-7Rα mAbs binding at site-1 or site 1/2b despite highly similar affinities, pharmacological behaviors and antagonist properties on STAT5 signaling pathway, led us to further characterize in vitro the effect of these two types of anti-IL-7Rα mAbs

on all previously described IL-7 signaling pathways and to further explore the biological impact of these mAbs on T lymphocytes. We identified another previously described site-1 mAb from others with similar affinity (clone 1A11 from WO/2011/094259 with a KD of $6 \times 10^{-10}$ M) that we recombinantly expressed with a human IgG1 Fc isotype as being developed in the clinic (NCT02293161). Analysis of conformational epitope using hydrogen deuterium exchange with mass spectrometry (HDX-MS) confirmed previous observations on linear epitopes and demonstrated that the site-1/2b mAb protected from deuterium incorporation in several peptides of the site-1 but also to a peptide overlapping the site-2b, while the two other mAbs (clone 1A11 and our site-1 IgG4 mAb) significantly prevented deuterium incorporation only in peptides from site-1 (Fig. 3a, b).

We then studied the impact of these confirmed site-1 and site-1/2b mAbs on STAT5, PI3K and ERK signaling pathways previously associated with IL-7R signaling. As expected, IL-7 induced potent STAT5 phosphorylation on human PBMCs and all three mAbs were potent inhibitors of IL-7-induced STAT5 phosphorylation (Fig. 3c, d, Supplementary Figure 6). In contrast, we found that while IL-7 induced a variable PI3K phosphorylation and does not induce ERK phosphorylation, the two site-1 mAbs significantly induced ERK and to a lesser extent PI3K signals even in the absence of exogenous IL-7 (Fig. 3c, d, Supplementary Figure 7). These findings show that these two site-1 mAbs have partial agonist properties and could be therefore considered as dual agonist/antagonist mAbs for the human IL-7R.

We assessed if these dual agonist/antagonist anti-IL-7Rα mAbs could deliver an effective agonist signal capable of modifying human T cell biology. We analyzed the transcriptome of human PBMCs incubated for 3.5 h with a site-1 (IgG4 #1 or IgG1 #2) or site-1/2b (IgG4) mAbs ± exogenous human IL-7 (5 ng/ml) by RNA sequencing (RNA-SEQ). Altogether, 481 genes were differentially expressed in human PBMCs incubated with anti-human IL-7Rα mAbs compared to control conditions. Sixty-one genes were differentially expressed with the three mAbs as compared to control, without any particular Gene ontology enrichment (Fig. 4a, left). Despite 61 common genes, the site-1/2b mAb induced only 31 significant gene modifications. In contrast, the two site-1 mAbs induced an important transcriptional modification of human PBMCs, with 245 and 237 differentially expressed genes induced by the site-1 IgG4#1 and the site-1 IgG1#2 mAbs respectively (78 genes were common between these 2 site-1 mAbs but not the site-1/2b mAb). Gene ontology enrichment suggested that both site-1 mAbs modified biological PBMC functions with genes related to leukocyte activation/differentiation (e.g., *OX40L, KCNA3, KDM6B*), proliferation/survival (e.g., *SIPA1L3, PIM3, PLK3*), migration/adhesion (e.g., *CD6, XLC2*), cytokine secretion and inflammatory responses associated with the MAPK/ERK pathway (Fig. 4a, bottom; Supplementary Figure 8). The gene modifications induced by the mAbs were strongly different from the modifications induced by IL-7 as displayed by the principal component analysis (PCA) of the IL-7 signature, i.e., 93 genes differentially expressed with IL-7 stimulation alone compared to medium control (Fig. 4a, right).

These 93 genes strongly differentially expressed (fold change > 2) with IL-7 compared to the control condition were separated into three distinct clusters by heat-map analysis (Fig. 4b, up). The first cluster of downregulated genes with IL-7 was composed mainly of genes implicated in leukocyte differentiation and apoptosis according to gene ontology enrichment, the third cluster of highly upregulated genes with IL-7 was associated with leukocyte adhesion, differentiation and activation, and a second cluster of upregulated genes had mixed gene ontology (Supplementary Figure 9). Interestingly, all mAbs significantly prevented

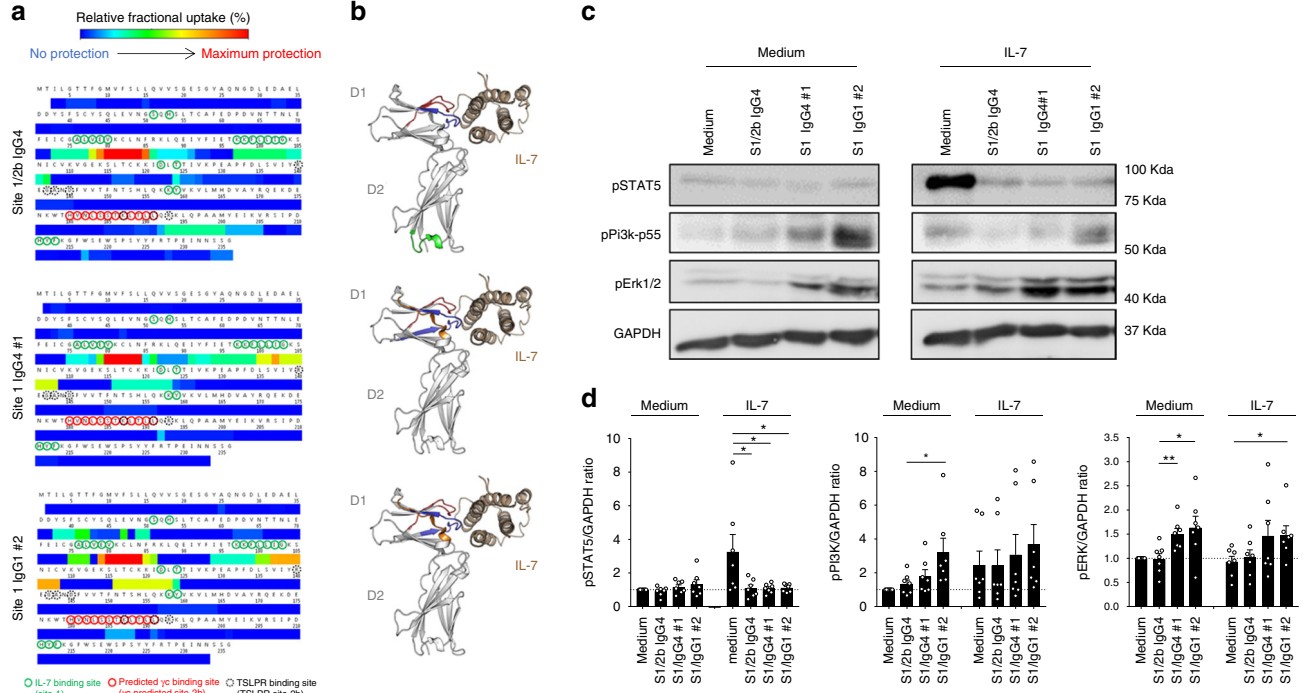

**Fig. 3** Anti-human IL-7Rα mAbs, epitopes, and antagonist/agonist signals. **a** Epitope determination by Hydrogen Deuterium Exchange and Mass Spectrometry (HDX-MS). Heatmaps of the relative fractional uptake differences between the bound and unbound antigen (recombinant human IL-7Rα) with the site-1/2b IgG4, site-1 IgG4 #1, and site-1 IgG1 #2 anti-human IL-7Rα mAbs. The relative fractional uptake of each amino acid is inferred from the value obtained for the smallest overlapping peptide, or the most C-terminal if two or more peptides are of equal length. Amino acids essential for IL-7 binding on IL-7Rα (site 1) are circled green, the predicted site-2b on IL-7Rα interaction with γ-chain is circled red and the interaction site of IL-7Rα with TSLPR circled in dotted black as previously described[17,19,59]. **b** Representation of the IL-7 (brown)/IL-7Rα (grey) structure of each epitope recognized by site-1/2b or site-1 anti-human IL-7Rα mAbs as determined by HDX-MS in (**a**). Epitopes (red, blue, orange or green colors) were determined on the most intense relative fractional uptake differences between the bound and unbound antigen. **c** Representative phospho-STAT5, phospho-PI3k-p55, phospho-ERK1/2, and GAPDH western blot of one out of seven representative human donor cells. PBMCs were pretreated with 10 μg/mL of one anti-IL-7Rα mAb and then incubated for 10 min at 37 °C with or without 5 ng/ml of human IL-7. **d** Same as (**c**) with quantification of pSTAT5, pPI3K, and pERK signals corrected to GAPDH expression and normalized to medium control conditions ($n = 7$ different donors). *$p < 0.05$; **$p < 0.01$ between indicated groups

cluster-1 and cluster-3 modification but without impact on the mixed cluster-2 (Fig. 4b, bottom). While site-1/2b mAb was more efficient in vivo, it sounds to be less efficient in vitro in preventing some IL-7-induced gene expression modification within the cluster-3, such as for example the anti-inflammatory *IKZF4* member of the Ikaros family of transcription factors, implicated in the control of lymphoid development. This result has been confirmed by RT-qPCR (Supplementary Figure 10) and suggests that some anti-inflammatory effect of IL-7 might be conserved by the site-1/2b mAb. Altogether, transcriptional analyses confirmed that while site-1 and site-1/2b anti-human IL-7Rα mAbs shared similar antagonist properties, the two site-1 mAbs induced significant transcriptional modifications of human PBMCs compatible with T-cell activation and inflammatory responses induced by the MAPK/ERK pathway.

**Anti-IL-7R induces antigen-specific memory T cell tolerance.** To further characterize in vivo the mechanism behind long-term control of memory T-cell mediated skin inflammation, we treated new BCG-vaccinated baboons with a humanized variant (CDR grafting into human antibody framework) of the "antagonist-only" (site-1/2b) anti-IL-7Rα IgG4 mAb (10 mg/kg, $n = 4$). In support of our previous findings, we found that three out of four baboons showed significant and long-term inhibition of the erythema response and epidermal thickening (a hallmark of skin inflammation) (Fig. 5a, b), as well as T-lymphocyte and macrophage skin infiltration (Fig. 6). Interestingly, while the number

and frequency of total memory T cells subsets remained unchanged after treatment (Fig. 7a), we found that the frequency of tuberculin-specific memory T cells (analyzed ex vivo by PBMC restimulation with tuberculin and numbers of IFN-γ-secreting cells by ELISPOT) significantly decreased after treatment and intradermal challenge with tuberculin (Fig. 5c). For an unexplained reason, one animal did not present inhibition of skin inflammation (overall, one out of ten animals treated with a "antagonist-only" anti-IL-7Rα mAb in Figs. 2 and 5). In parallel, this animal did not display the decrease in IFN-γ-secreting cells ex vivo after tuberculin restimulation that had been observed in other responding animals (Supplementary Figure 11).

The frequency of tuberculin-specific IFN-γ+ memory T cells in the responders remained significantly decreased overtime in parallel with long-term inhibition of erythema, epidermal thickening and immune cell skin infiltration (Figs. 5 and 6). Addition of exogenous IL-2 or depletion of CD25+ cells from PBMCs prior to restimulation in this assay failed to implicate these factors as potential drivers of hyporesponsiveness suggesting that T-cell anergy and Tregs are not responsible for the observed tolerance (Fig. 5d). As previously observed for the first six animals treated with site-1/2b anti-IL-7Rα mAb (Fig. 2c), tuberculin-induced skin inflammation was visible again in these three new responders after BCG re-vaccination (Figs. 5a, b and 6) in parallel with a recovered frequency of peripheral tuberculin-specific memory T cells (Fig. 5c).

To decipher how anti-IL-7Rα mAb might induce selective antigen-specific memory T cell elimination in vivo, we first

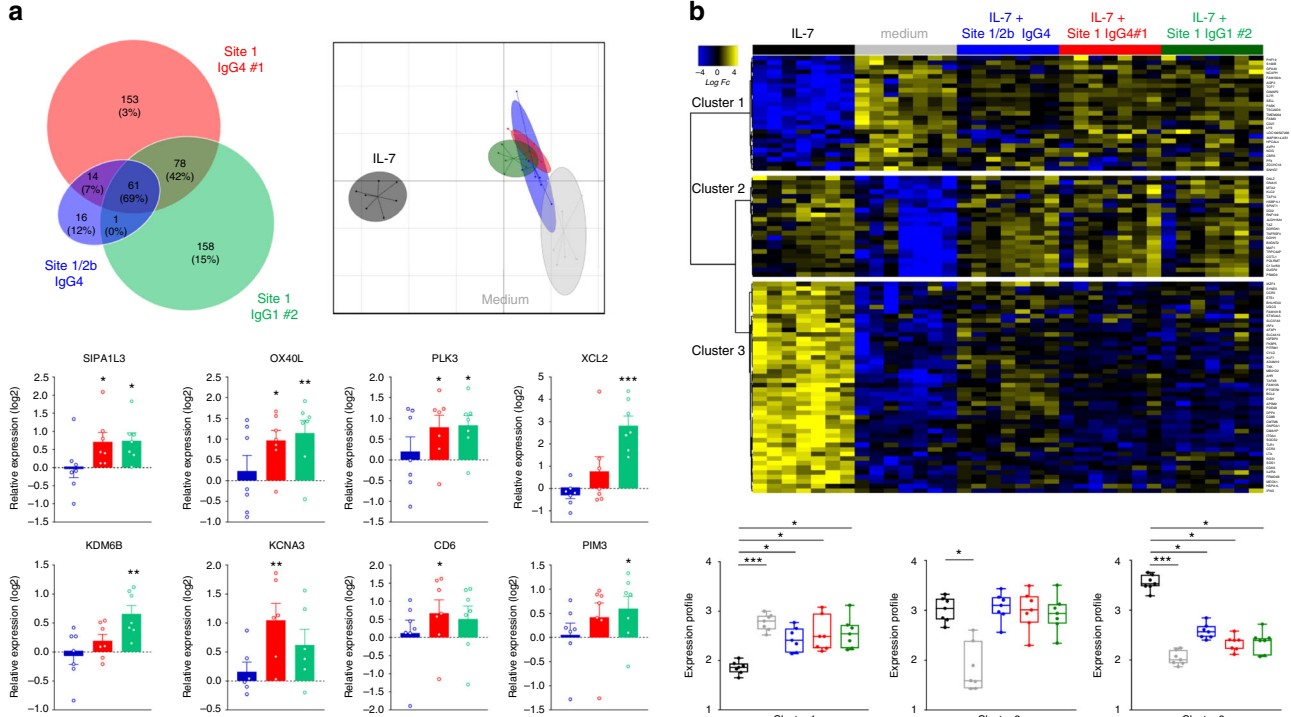

**Fig. 4** Dual agonist/antagonist (site-1) anti-IL-7Rα mAbs induce transcriptional modification and activation of human leukocytes. **a** RNA-Seq analysis of human PBMCs ($n = 7$) incubated without IL-7 for 3.5 h with different anti-human IL-7Rα mAbs (10 μg/ml, blue: site-1/2b IgG4, red: site-1 IgG4#1, green: site-1 IgG1#2). Upper left: Venn diagram of the 481 differentially expressed genes (FDR 5%, FC > 1.5) comparing anti-IL-7Rα mAbs and medium control conditions. Upper right: Principal component analysis (PCA) of anti-IL-7Rα mAbs versus control and IL-7 stimulation (5 ng/ml) on the 93 most differentially expressed genes (FDR 5%, FC > 2) comparing IL-7 stimulation and control conditions. Bottom: Relative expression (fold-change as compare to control medium condition) of selected genes with each anti-IL-7Rα mAb (blue: site-1/2b IgG4, red: site-1 IgG4#1; green: site-1 IgG1#2). *$p < 0.05$; **$p < 0.01$; ***$p < 0.001$ as compare to control medium condition. **b** RNA-Seq analysis of human PBMCs ($n = 7$) incubated with IL-7 (5 ng/ml) for 3.5 h with the different anti-human IL-7Rα mAbs (10 μg/ml). Upper: Heatmap of the expression of the 93 most differentially expressed genes (FDR 5%, FC > 2) between IL-7 stimulation and control condition. Bottom: Quantification of the median profile of the three IL-7 induced clusters in IL-7 stimulated, control and IL-7 + anti-human IL-7Rα mAbs conditions. Same colors as in (**a**). *$p < 0.05$; **$p < 0.01$; ***$p < 0.001$ between indicated groups

evaluated the impact of IL7 in vitro on the proliferation and survival of human CD8+ and CD4+ memory T cells after restimulation with MHCI (H1H1 flu) and MHCII (CMV, EBV, Flu, Tetanos) -restricted peptides, respectively, and compared with polyclonal stimulation. While IL7 had no significant impact on the proliferation of human memory CD4 or CD8 T cells after 3 days of culture and only a modest (non-significant) positive effect on memory T cells survival (Fig. 8a), at late time-point (after a week of stimulation), IL-7 significantly and strongly improved memory CD4 and CD8 human T cells survival after antigen restimulation (Fig. 8b). Blocking IL-7/IL-7R axis using anti-IL-7Rα mAb prevented the beneficial effect of IL-7 on memory T cells proliferation and survival (Fig. 8c). Interestingly, no significant impact was found on human T cells after polyclonal stimulation while cells proliferated vigorously and displayed very weak survival after 8 or 10 days of stimulation (Fig. 8b, c). Interestingly, IL7 increased survival only in proliferating human cells (Fig. 8c middle) while in the same wells IL7 had no impact on quiescent cells (Fig. 8c right). Altogether, the IL7/IL7R axis sustains the survival of memory T cells after antigenic rechallenge and anti-IL7Ra mAbs prevent selectively the survival of antigen-specific memory T cells after restimulation.

To prove that antagonist anti-IL-7Rα mAbs induced antigen-specific hyporesponsiveness and not broader immunodeficiency, we first assessed ex vivo polyclonal activation of baboon PBMCs before and during treatment. T lymphocytes from treated animals exhibited very low levels of CD25 and CD69 activation markers

without stimulation. In contrast, they dramatically overexpressed these activation markers after polyclonal activation with or without exogenous IL-2 or IL-7 (Fig. 7b, Supplementary Figure 12). No significant difference was observed after treatment compared to pretreatment activation status. The metabolic profile of the pool of peripheral T lymphocytes of treated animals also remained unchanged after treatment since T cells were still able to increase glycolytic activity (measured by extracellular acidification rate: ECAR) and mitochondrial respiration (measured by the mitochondrial oxygen consumption rate: OCR) after polyclonal ex vivo stimulation (Fig. 7c). Similarly, intracellular ATP content and mobilization in response to stimulation as well as glucose uptake (measured by incorporation of the 2-NBDG fluorescent glucose analog) were also unmodified after restimulation ex vivo (Fig. 7d). Altogether, these results strongly suggest that antagonist anti-IL-7Rα mAbs do not provoke general immunosuppression but induce long-term immune tolerance in non-human primates through clonal deletion of antigen-specific memory T cells after antigen rechallenge.

## Discussion

The identification of alternative therapeutic approaches targeting more upstream mechanisms in autoimmune and chronic inflammatory diseases is desired for preventing relapse and maintaining long-term remission in patients. Whilst the interest for the homeostatic T cell proliferation pathway is increasing[48], there is still a substantial lack of molecules targeting this pathway

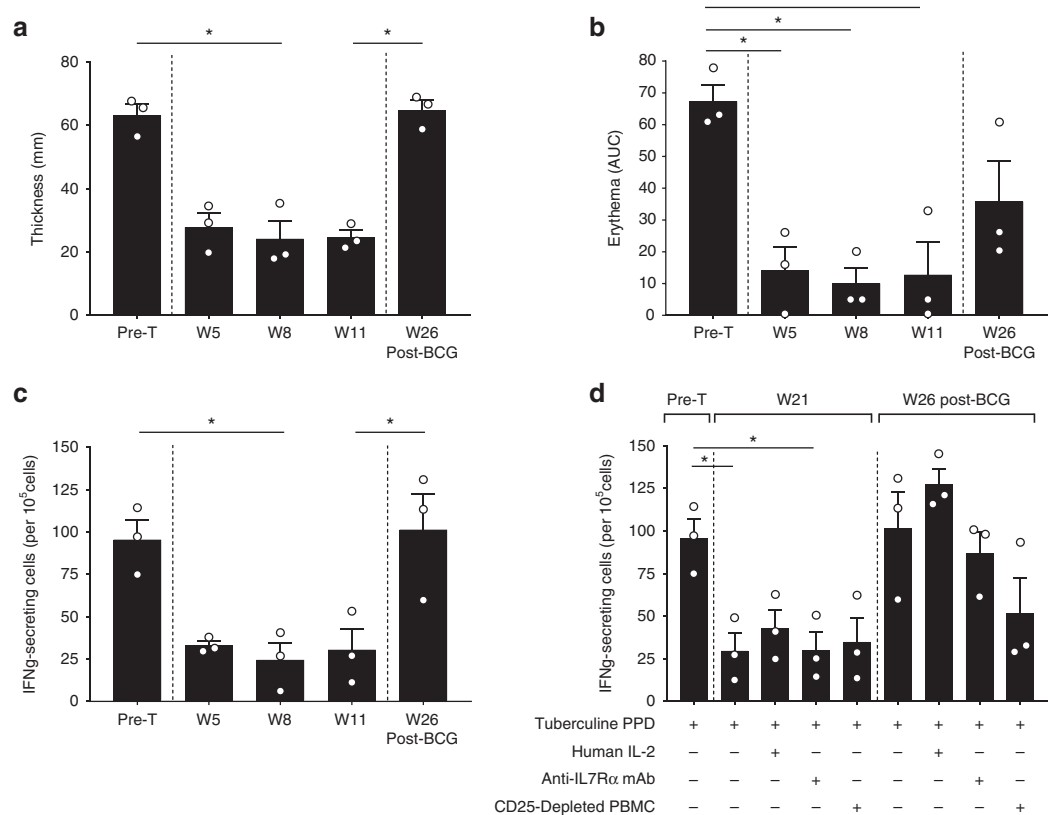

**Fig. 5** Antagonist anti-IL-7Rα mAb induces antigen-specific memory T cell hyporesponsiveness. **a** Epidermal thickness measured histologically on skin biopsies performed 4 days after tuberculin intradermal injection in baboons treated with a single intravenous injection of 10 mg/kg of a humanized site-1/2b IgG4 mAb (n = 3). Pre-T: 4 weeks pre-treatment with the mAb; W5, W8, W11 are the number of weeks (W) after mAb administration. W26 post-BCG is a new tuberculin challenge after re-vaccination of animals with BCG vaccine. **b** Cutaneous erythema response at the indicated time-point as in (**a**), represented with area under the curve (AUC) of daily erythema diameters. **c** IFN-γ-secreting cell frequencies in baboon PBMCs after ex vivo tuberculin restimulation at the indicated time-point as in (**a**). **d** At indicated time-points, week 21 (before BCG re-vaccination) and week 26 (post-BCG re-vaccination), 600 IU/ml of recombinant human IL-2 or 10 μg/mL of the humanized site-1/2b IgG4 mAb were added in some wells during ex vivo tuberculin restimulation. In some conditions, CD25⁺ depletion was performed on PBMCs before tuberculin restimulation to assess a potential role of Tregs. Data are mean ± SEM and erythema diameters were measured by at least two observers. *$p < 0.05$; **$p < 0.01$ one-way ANOVA with Dunn's multiple comparison test

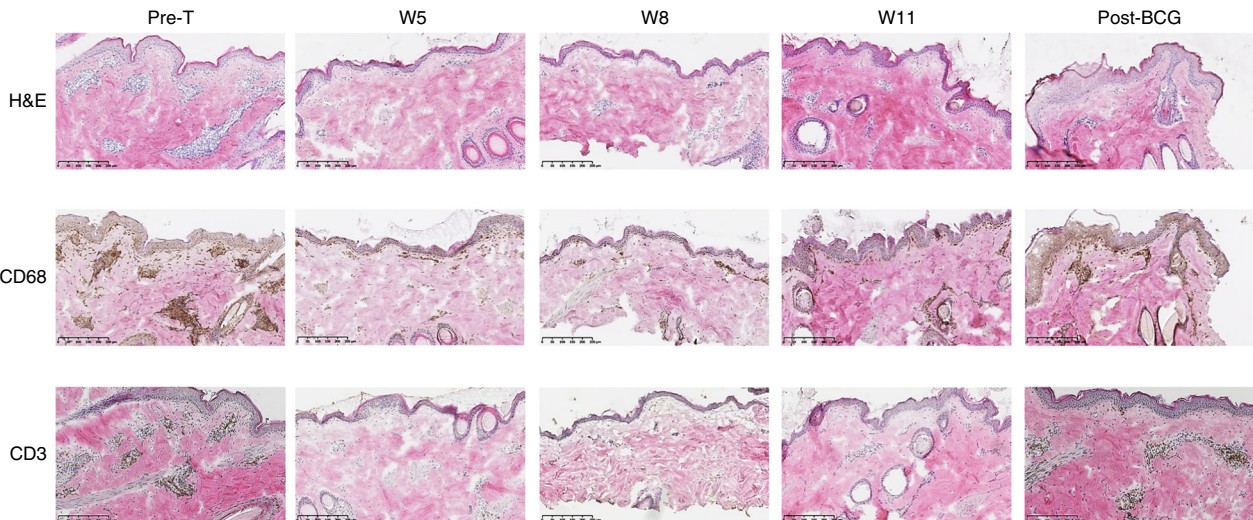

**Fig. 6** Antagonist anti-IL-7Rα mAb inhibits immune cells skin infiltration. Representative hematoxylin & eosin (upper), CD68 (middle), and CD3 (bottom) staining of skin biopsies performed 4 days after tuberculin challenge at the indicated time-points in animals treated with a single intravenous injection of 10 mg/kg of a humanized site-1/2b IgG4 mAb. Pre-T: 4 weeks pre-treatment with the mAb; W5, W8, W11 are numbers of weeks (W) after mAb administration. Post-BCG: new tuberculin challenge after re-vaccination of animals with BCG

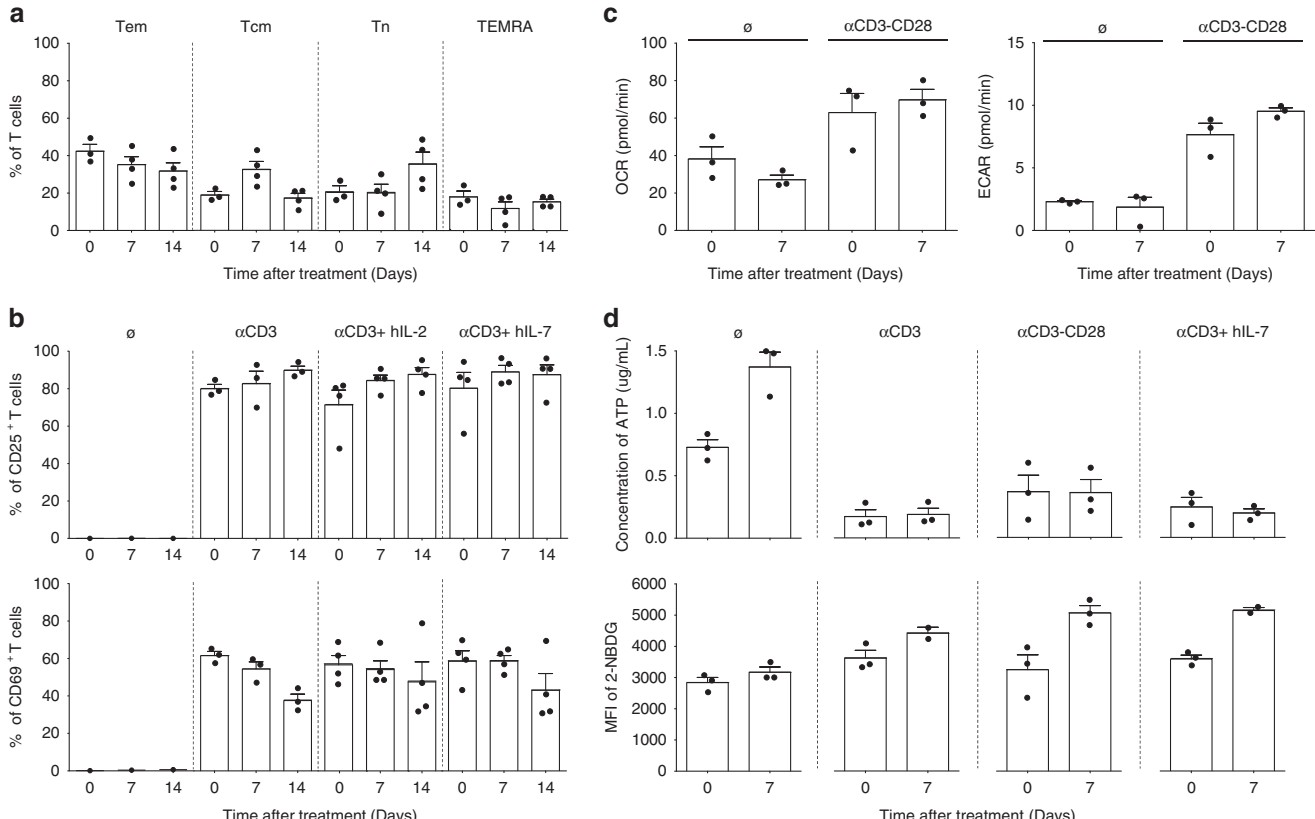

**Fig. 7** Antagonist anti-IL-7Rα mAb does not disturb polyclonal T-cell activation or modify metabolism. **a** Peripheral blood T-cell subset frequencies determined by flow cytometry for baboons treated with a single intravenous injection of 10 mg/kg of a humanized site-1/2b IgG4 mAb ($n = 4$). T cell sub-populations were defined using the following gating strategy CD3$^+$ cells for Tem: effector memory T cells (CCR7$^-$ CD45RA$^-$), Tcm: central memory T cells (CCR7$^+$ CD45RA$^-$), Tn: naive T cells (CCR7$^+$ CD45RA$^+$), TEMRA: CD45RA expressing effector memory T cells (CCR7$^-$ CD45RA$^+$). **b** CD25$^+$ (upper) and CD69$^+$ (lower) activated T-cell frequencies at the indicated time-point in the same animals as in (**a**), after ex vivo culture (48 h), without stimulation or with anti-CD3 polyclonal stimulation, and supplemented with human IL-2 (300 UI/mL) or human IL-7 (10 ng/mL) as mentioned. **c** Oxygen consumption rate (OCR) and extracellular acidification rate (EACR) of T cells purified one week before and after anti-IL-7Rα treatment in the same animals as in (**a**). OCR and EACR was determined on freshly isolated and unstimulated T cells or after over-night polyclonal activation using Seahorse XF24 Analyzer. **d** ATP concentration (μg/mL) and mean fluorescence intensity (MFI) of 2-NBD glucose uptake in the same condition as in (**c**). Data are mean ± SEM

in humans and a lack of relevant preclinical trials to assess whether this could be an effective approach to control memory T cell-mediated autoimmunity and chronic inflammation. It has been shown previously that blocking the IL-7R pathway can reverse ongoing autoimmunity[39] or inhibit allograft rejection[37,38] in rodent models. However, it has generally been difficult to delineate the exact mechanisms due to T cell lymphopenia observed in most studies performed in rodents and the poor description of mAbs properties. We found that the agonist/antagonist properties of anti-IL-7Rα mAbs depends on the specific epitope targeted, since antibodies binding the IL-7 interaction domain (site-1) appear to have both agonist/antagonist properties whereas an antibody binding in addition to the heterodimerization domain of IL-7Rα/γc (site-2b) displays strict antagonist activity. Although we have not been able to demonstrate the capacity of this "antagonist-only" antibody to modify the IL-7R complex, we propose that it could perturb IL-7Rα/γc dimerization required for receptor internalization and signaling. "Antagonist-only" antibodies against the IL-7R prevented long-term memory T-cell-mediated skin inflammation in primates, even after chronic antigen stimulation, without inducing lymphopenia or polyclonal T-cell functional or metabolic deficiencies. Our data show that blocking IL-7R during antigen recall selectively abrogated the response of antigen-specific memory T

lymphocytes in a long-term manner, while a secondary cycle of vaccination could revert this antigen-specific tolerance, suggesting a selective elimination mechanism.

IL-7 has been shown to induce proliferative and anti-apoptotic signals through IL-7R signaling mainly by activating the JAK/STAT pathway. IL-7R signaling is also believed to involve the PI3K/AKT pathway, but this has been observed in transformed immortalized cell lines or primary thymocytes and these signals were not detectable in peripheral naive or memory human T lymphocytes[20]. Several reports have also suggested that IL-7R signaling could amplify ERK phosphorylation either in T lymphocytes or pro-B cell subsets[49,50]. We confirmed that IL-7 induced reproducible STAT5 phosphorylation, PI3K signal activation was more variable, and we did not observe any ERK phosphorylation. While all anti-IL-7Rα mAbs used in this study were potent inhibitors of IL-7-induced pSTAT5 and displayed similar transcriptional antagonist properties, we found that two site-1 mAbs induced PI3K/ERK agonist signals and important transcriptional modifications associated with T-cell activation and inflammatory responses induced by the MAPK/ERK pathway. On the other hand, some IL-7-induced anti-inflammatory gene overexpression were efficiently inhibited with site-1 mAbs but not with the site-1/2b mAb (see heatmap in Fig. 4b). For example, the Ikaros family member *IKZF4* which is highly induced by IL-7 and

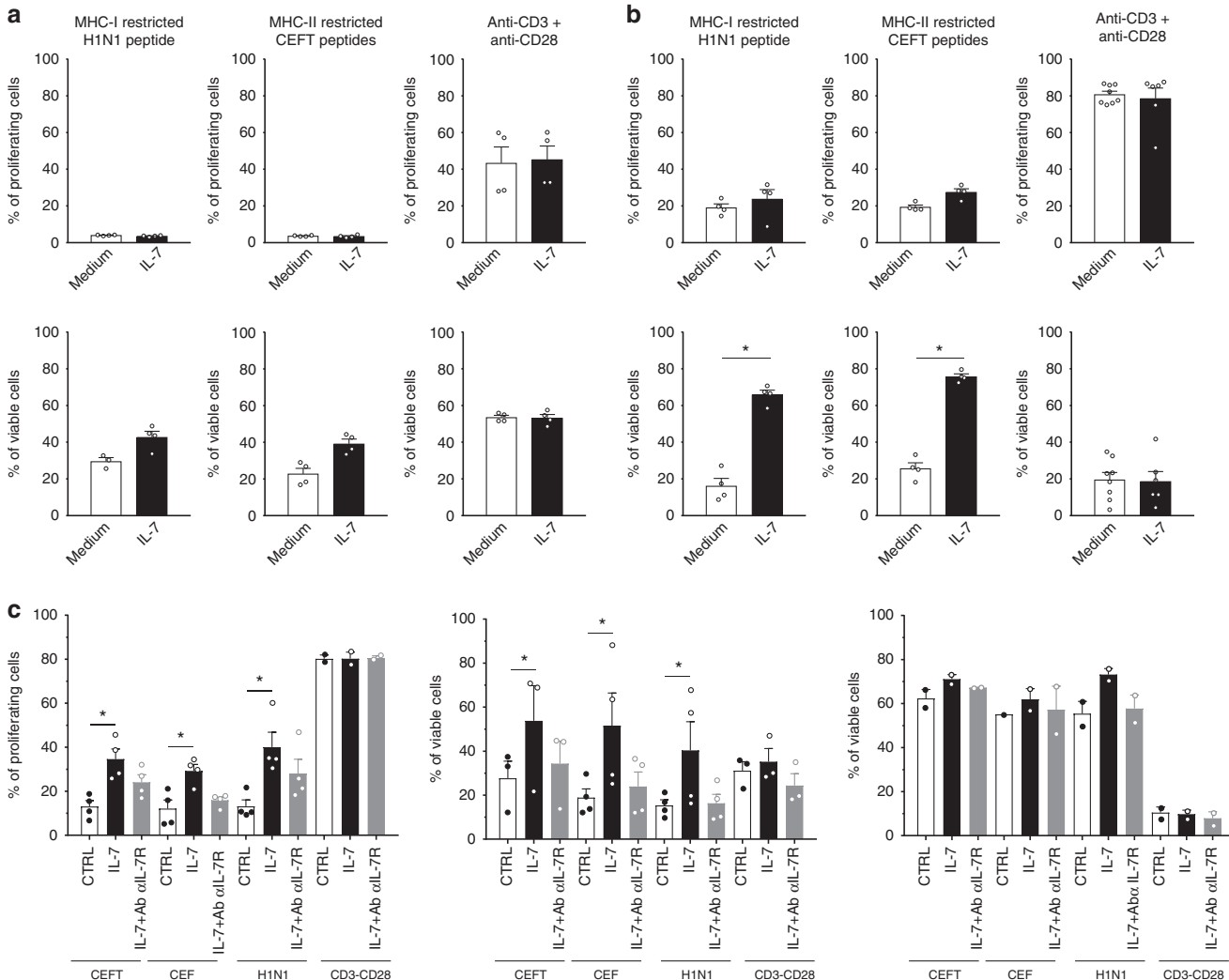

**Fig. 8** Antagonist anti-IL-7Rα mAb inhibits antigen-specific human memory T cells persistence after antigen rechallenge. Human CPD-labeled PBMCs ($n = 4$) were stimulated with MHCI or MHCII -restricted pool of peptides (H1N1 flu; CEFT: CMV, EBV, Flu, Tetanos; CEF: CMV, EBV, Flu) or with anti-CD3 + anti-CD28 mAbs for 3 days (**a**), 8 days (**b**) or 10 days (**c**). Cells were cultured in medium alone (white histogram) or supplemented with 5 ng/ml of recombinant human IL-7 (black histogram) or 5 ng/mL of human IL-7 plus 10 µg/ml of the site-1/2b humanized anti-IL-7Rα mAb (grey histogram). The histograms show the percentage of proliferated cells (% of CPD$^{low}$ cells) and viable cells (% of Annexin-V$^−$ PI$^−$ cells) among total cells, or within proliferated (CPDlow, Fig. 7c middle) versus quiescent (CPD high, Fig. 7c right) cells. Horizontal bars mean ± SEM. *$p < 0.05$ between indicated groups

not clearly affected by site 1/2b Ab, was reported to prevent TH17 polarization[51], therefore it might be conceivable that some anti-inflammatory actions of IL-7 are differentially inhibited by the two classes of anti-IL-7R mAbs and might contribute to the difference observed in vivo. These opposing dual agonist/antagonist properties of some mAbs are not unique since other targets such as IL-4[52], IL-6R[53], IL-15[54], CD28[55], CD38[56], CD40[57], or HER2 demonstrated similar activities after receptor endocytosis/internalization[58]. The absence of agonist signals with an anti-IL-7Rα mAb was found with an antibody targeting an epitope overlapping to the predicted site of heterodimerization (site 2b) between the alpha and gamma chain in structural studies[19]. The heterodimerization of this site 2b with TSLPR has also been recently confirmed and demonstrated to play a role in receptor signaling, as already predicted for IL-7Rα and the γ-chain[59]. However, while IL-7Rα employs similar, but not identical, set of residues at site-1 to interact with IL-7 and TSLP[59], neither site-1 nor site1/2b anti-IL-7Rα mAbs displayed significant inhibition of TSLP-induced TARC secretion by human dendritic cells.

We found that the in vitro pSTAT5 inhibition assay was not predictive of anti-IL-7R antibody efficacy in vivo since our dual agonist/antagonist (site-1) mAb did not control T cell responses in primates whereas "antagonist-only" (site-1/2b) anti-IL-7Rα mAbs, independently of isotype, inhibited memory T cell-induced skin inflammation. Yet both types of antibodies had similar potency for inhibiting STAT5 phosphorylation in vitro while binding with superimposable pharmacodynamic properties after i.v. administration. Similarly, in a previous report, another anti-IL-7Rα mAb prevented in vitro and ex vivo IL-7-induced pSTAT5 in primates but did not protect from brain inflammation in an experimental autoimmune encephalitis (EAE) marmoset model[42]. In line with our site-1 specific mAb in vivo results in baboons (Supplementary Figure 2A), the same group described in a different report that their anti-IL-7Rα mAb was able to induce IL-7Rα internalization/downregulation in cynomolgus monkeys[47].

In contrast to most studies performed in rodent models where IL-7R blockade induce broad lymphodepletion[37], administration of high doses (i.e., 10 mg/Kg) of anti-IL-7Rα mAb targeting either

site-1 or site 1/2b in non-human primates (here in baboons or previously by others in marmoset[42] or cynomolgus monkey[47]) did not induce lymphopenia or significant decrease in peripheral T lymphocyte numbers. Similarly, early clinical results from phase 1 trials with two different site-1 anti-IL-7Rα mAbs in healthy volunteers (GSK2618960, NCT02293161) or in type 1 diabetic adult patients (PF-06342674, NCT02038764) did not report induction of lymphopenia (https://www.gsk-clinicalstudyregister.com/files2/gsk-200902-clinical-study-result-summary.pdf)[60].

Thus, if IL-7 has been well-described in maintaining the pool of peripheral naive and memory T lymphocytes in mice, in primates, however, the importance of IL-7 in maintaining peripheral T-cell homeostasis is less evident and/or redundant mechanisms might explain the difference between species. For example, CD8+ T cells can use both IL-15 and IL-7 for their maintenance in the periphery[61] or tissues[26], suggesting that the absolute requirement for IL-7 is questionable. This is in contrast with the importance of IL-7 for T-cell development where IL-7R deficient/mutated mice and humans demonstrate severe T lymphocyte deficiency[62].

A single administration of an "antagonist-only" anti-IL-7Rα mAb induced a very long-term (at least up to 1 year) inhibition of skin inflammation mediated by chronic antigen recall and memory T lymphocyte reactivation. In our study, animals were generally not immunocompromised because they were able to elicit immune responses in vivo to SRBC immunization after anti-IL-7Rα mAb elimination. Quiescent naive and memory T cells use β-oxidation of fatty acids for self-sustenance but switch to glycolysis after activation to support rapid growth in response to antigen recognition[63]. IL-7 has been shown to promote glycolysis in TCR-activated T cells[64]. Here, polyclonal activation and metabolic responses of T cells from these animals were not altered, confirming an absence of polyclonal T cell defects following anti-IL-7Rα therapy in primates which differs from previous mouse studies.

Due to the absence of available MHC tetramer molecules specific for baboons, the frequency of antigen-specific memory T cells was measured by ex vivo PBMC restimulation using tuberculin and IFN-γ ELISPOT assays. We found that ex vivo tuberculin-specific IFN-γ-secreting memory T cells significantly decreased after anti-IL-7Rα mAb therapy and perfectly correlated with the inhibition of skin inflammation otherwise induced by antigen challenges. However, ELISPOT assays could be biased by intrinsic or extrinsic mechanisms of immune tolerance preventing IFN-γ secretion. While some reports demonstrated a key role of Tregs in the protective mechanisms induced by IL-7R blockade in rodent models[37,38], here the depletion of CD25+ cells from PBMCs did not restore IFN-γ secretion. Others have described the intrinsic control of memory T-cells after IL-7R blockade in rodent models of type 1 diabetes through upregulation of PD-1 coinhibitory molecules[27,28]. Here, we did not observe increased PD-1 expression on peripheral blood memory T cells and the activation/metabolism of T cells in response to polyclonal stimulation was not modified, suggesting that T cells are not exhausted or inhibited by intrinsic factors. Similarly, addition of high concentration of IL-2 in ELISPOT assays did not restore IFN-γ responses excluding an anergic state of antigen-specific memory T cells which could not be detected by polyclonal stimulation. Altogether, these results argue more simply for the disappearance of antigen-specific memory T cells from the blood (and the skin) of animals after anti-IL-7Rα mAb treatment through clonal deletion mechanisms, explaining the antigen-specific desensitization of the animals. A recent report in a chronic EAE model in mice supports this. Indeed, it showed that short-term anti-IL-7Rα mAb treatment inhibited the expansion of autoantigen-specific T cells by inhibiting recently activated autoreactive T cells leading to selective and major (>90%)

depletion of antigen-specific memory T cells[30]. Using TCR-transgenic mice and adoptive transfer experiments, the authors identified antigen-specific pre-apoptotic cells at early time-points, in accordance with the significant role of IL-7 in Bcl-2/Bcl-xL regulation after antigen reactivation. Indeed, we found in vitro that IL-7 controls selectively the survival of proliferating (not quiescent) human memory T cells after antigenic restimulation and not polyclonal stimulation. Altogether, anti-IL-7Rα mAb induces selective elimination of antigen-specific memory T cells after antigen challenge by preventing this anti-apoptotic IL-7/IL-7R axis required by memory T cells to persist after restimulation. Finally, we found that animals re-vaccinated long after drug withdrawal recovered a normal immune response, in accordance with the mechanism of action of antigen-specific memory T cells clonal deletion upon treatment which should not affect de novo priming of naive T cells particularly after drug elimination. Hence, while anti-IL-7Rα mAb treatment demonstrated pre-clinical efficacy, preventing relapse in patients suffering from chronic inflammatory and/or autoimmune diseases will probably require iterative administrations. Furthermore, while therapeutic strategies inducing robust immune tolerance might suffer from their safety profile (becoming tolerant to an unwanted pathogen), here we showed that re-vaccination with live-attenuated pathogen was sufficient to generate de novo antigen-specific immune responses.

Our study shows that antagonist properties of anti-IL-7Rα mAbs and in vivo efficacy are not only related to the prevention of IL-7 binding and pSTAT5 inhibition. We suggested that antibodies also targeting the receptor heterodimerization site, such as those described here, are "antagonist-only" and result in a higher efficacy of inhibitory T cell responses in vivo. Presumably, residual signaling through receptor heterodimerization might prevent control of memory T cell reactivity, and our data reinforce the concept that antigen-engaged T-cell clones at early stages of activation are very sensitive to IL-7 withdrawal. Anti-IL-7Rα mAb induced antigen-specific tolerance most probably through elimination of engaged memory T cells during the period of treatment, with minimal effects on naive or memory T cell pools. Targeting IL-7R with "antagonist-only" antibodies has the potential to regulate antigen-specific memory T cell survival and accumulation, and therefore might promote the prevention of long-term relapse in autoimmune and inflammatory diseases.

## Methods

**Anti-IL-7Ra mAbs**. Anti-human IL-7Ra mAbs were generated by CDR-grafting technology after sequencing of hybridoma generated from Lou/C rats immunized with recombinant human IL-7Ra protein. CDR were grafted in most appropriate human VH/VL frameworks and combined with a human Fc domain from IgG4 (S228P mutation preventing Fab-arm exchange) or IgG1 (E333A mutation increasing FcγR affinity and cytotoxic functions) isotypes. Resulting humanized antibodies were then recombinantly expressed in HEK cells. The site-1 anti-IL-7Ra mAb clone 1A11, of the human IgG1 isotype, was identified in the literature (WO/2011/09259) and has been similarly recombinantly expressed in HEK cells. Antibodies have been purified from cell supernatant by a capture chromatography on Protein-A-Sepharose. Purity by HPLC was >90%.

**Animals**. Baboons (Papio anubis; 7–15 kg) were obtained from the Centre National de la Recherche Scientifique Centre de Primatologie (Rousset, France). The animals were housed at the large animal facility of the INSERM UMR 1064. All experiments were performed in accordance with our institutional guidelines and approved by the French National Ethics Committee (no. CEEA-855).

**Western blotting**. Freshly isolated human PBMCs from healthy donors (obtained by the Establissement Français du Sang) were incubated for 30 min at 37 °C with 10 μg/ml of anti-human IL-7Rα mAbs, and then cultured alone or with 5 ng/ml of recombinant human IL-7 (AbDSerotec) for 10 min at 37 °C. After stopping the reactions on ice, cells lysates were prepared with RIPA buffer (with Protease Inhibitor Cocktail). Proteins (15 μg) were resolved under reducing conditions on 7.5% polyacrylamide gels and immobilized on nitrocellulose membranes (GeHealthcare) using standard methods. Blots were washed with 5% BSA-tris

buffer saline and incubated with Phospho-STAT 5, Phospho-PI3K p55 or Phospho-ERK specific antibodies in 1% BSA-TBS (overnight at 4 °C), followed by a polyclonal goat anti-rabbit horseradish peroxidase-labeled antibody (Cell Signalling Technology) for 1 h at room temperature. Alternatively, blots were stained using a GAPDH antibody (Santa Cruz) in 1% BSA-TBS (overnight at 4 °C), followed by polyclonal goat anti-mouse horseradish peroxidase-labeled antibody (Jackson Immunoresearch) for 1 h at room temperature. Membranes were revealed by chemiluminescence using a LAS-3000 imaging system (Fujifilm).

**Epitope determination.** Using the HDX-2 system (Waters S.A./N.V.; Zellik, Belgium), recombinant human CD127 and 0 or 1 molar equivalent of mAb were mixed and diluted in 99.9% $D_2O$, 10 mM sodium phosphate, 100 mM NaCl, pH 6.8 to a final $D_2O$ content of 90% and a CD127 or CD127/mAb complex concentration of 27.5 μM. Hydrogen-deuterium exchange was performed at 20.0 °C for 30 min. The exchange was quenched by a 1:1 (v/v) dilution of samples with 100 mM sodium phosphate, 4 M guanidine·HCl, 0.4 M TCEP, pH 2.3, at 1.0 °C resulting in a final pH of 2.5. After 2 min, the quenched samples were loaded onto the HDX manager for online pepsin digestion at 20.0 °C (Enzymate BEH Pepsin, 2.1 × 30 mm; 5 μm), followed by desalting (Acquity BEH C18 Vanguard 2.1 mm × 5 mm; 1.7 μm) and reverse phase separation (Acquity BEH C18 1.0 mm × 100 mm; 1.7 μM) using a gradient from 5% to 40% 0.2% formic acid in acetonitrile (pH 2.5) for 10 min at a flow rate of 40 μl/min at 0.0 °C. Mass spectrometry analysis was performed on a Waters Xevo G2-XS ESI-Q-TOF mass spectrometer in the positive ion mode, with lockspray correction. Mild source conditions (temperature: 90 °C, capillary voltage: 2.5 kV, sampling cone: 30 V, desolvation gas flow: 800 L/h, desolvation temperature: 250 °C) were used in order to minimize back-exchange while ensuring proper desolvation[65]. Peptide identification was assisted by collision induced dissociation collected in the MSE mode, using PLGS 3.0.2 and UNIFI 1.8. Deuterium incorporation was determined in DynamX 3.0. Structural figures were prepared using PyMOL 1.8.2.3 (Schrödinger LLC, Cambridge, MA, USA) from PDB ID 3DI3[17]. Monobasic and dibasic sodium phosphate, sodium chloride, guanidine hydrochloride, Tris(2-carboxyethyl)phosphine hydrochloride (TCEP), 50% sodium hydroxide, and formic acid were purchased from Sigma Aldrich (Schnelldorf, Germany) at the highest available purity. LC-MS grade solvents were sourced from Biosolve Chimie (Dieuze, France), deuterium oxide (99.9% D) and 20% deuterium chloride in deuterium oxide (99.96% D) from Cambridge Isotope Laboratories (Andover, MA, USA), hydrochloric acid 37% from VWR International (Fontenay-sous-Bois, France), and bovine cytochrome C digest from Thermo Fisher Scientific (Germering, Germany). Amicon ultra-centrifugal filters (0.5 mL; 10 kDa cut-off) were obtained from Merck Millipore (Molsheim, France).

**RNA sequencing.** Freshly isolated human PBMCs from healthy donors (obtained by the Establissement Français du Sang) were incubated with 10 μg/ml anti-human IL-7Rα mAbs (30 min at 37 °C), and then cultured alone or with 5 ng/ml of recombinant human IL-7 (AbDSerotec) for 3 h at 37 °C. Reactions were stopped on ice and the cell pellets resuspended in RLT buffer (Qiagen) containing 1% β mercaptoethanol in RNase/DNase free water and stored at −80 °C. RNA was extracted using an RNA mini extraction kit according to manufacturer's instructions (Qiagen). The quality and quantity of RNA were assessed by infrared spectrometry (Nanodrop) and Agilent bioanalyzer (Agilent RNA 6000 Pico Kit). Smart-Seq2 libraries were prepared by the Broad Technology Labs and sequenced by the Broad Genomics Platform according to the SmartSeq2 protocol with some modifications[66]. Briefly, total RNA was purified using RNA-SPRI beads, polyA+ mRNA was reverse-transcribed to cDNA, and amplified cDNA was subject to transposon-based fragmentation that used dual-indexing to barcode each fragment of each converted transcript with a combination of barcodes specific to each sample. Sequencing was carried out as paired-end 2 × 25 bp with an additional 8 cycles for each index. Data was separated by barcode and aligned using Tophat version 2.0.10 with default settings. Transcripts were quantified by the Broad Technology Labs computational pipeline using Cuffquant version 2.2.1[67]. Briefly, data were processed through CuffNorm if 50% of the reads aligned, and if at least 100,000 pairs were aligned per sample. Normalization used the default settings, including "geometric" normalization, and expression level information as log2-transformed FPKM values (Fragments per kilobase of transcript per million mapped fragments) were used for subsequent analyses. For identification of differential genes, linear modeling with estimation of the mean-variance relationship (limma-trend) with empirical Bayes statistical procedure were performed using the *limma* package in R[68]. Genes with Benjamini and Hochberg adjusted *p*-value < 5% and fold change (FC) > 1.5 were considered as differentially expressed. For gene expression representation, PCA and clustering were performed in R v3.3.2 using *ade4/adegraphics* and *pheatmap* packages respectively. The biological significance of selected genes was assessed using the R *clusterProfiler* package. Gene ontology (GO) categories enriched with a false discovery rate (FDR) < 5% and with at least five represented genes were selected. RNA-seq data have been deposited in GEO under the accession code GSE103643.

**Tuberculin-induced DTH model.** Baboons were immunized intradermally twice with a BCG vaccine (0.1 ml; 2–8 3 105 CFU; Sanofi Pasteur MSD, Lyon, France) in the upper region of the leg, 4 and 2 weeks before the DTH skin test. Intradermal

reactions (IDR) were performed with duplicate 0.1 ml intradermal injections of two doses (2000 or 1000 IU) of tuberculin-purified protein derivative (PPD; Symbiotics Corporation, San Diego, CA, USA) in the skin on the right back of the animals. Saline injections (0.1 ml) was used as a negative control. Dermal responses at the injection sites were measured using a caliper square. The diameter of each indurated erythema was measured by at least two observers from day 3 to 12, and was considered positive when above 4 mm in diameter. The mean of the reading was recorded and plotted for each time point. To compare multiple experimental conditions, erythema responses were quantified as area under the curve (AUC) using Graph Pad Prism software for calculation. Skin biopsies were performed on day 4 on one duplicate. A second IDR was performed after 3 weeks and animals received one intravenous injection of one anti-human IL-7Rα mAb (10 mg/Kg) 24 h before the second challenge with tuberculin. Other IDR were performed every 3–4 weeks without any further injection of the drug. At late time-points, some animals were re-vaccinated with BCG vaccine following the same protocol as described above. Blood samples were drawn at several time-points for flow cytometric or metabolic analysis, serum cytokine analyses (BD CBA non-human primate Th1/Th2 Cytokine Kit, BD Bioscience) and pharmacokinetic analysis on sera by ELISA. Recombinant hCD127 (Sino Biologicals, Beijing) was coated on plastic at 1 μg/ml and dilutions of baboon sera were added to detect free anti-human IL-7Rα mAb. After incubation and washing, mouse anti-human light chain (kappa specific) plus peroxidase-labeled donkey anti-mouse antibodies were added and revealed by conventional methods.

**SRBC immunization.** Some baboons received an intravenous administration of 1.5 ml/kg of SRBC at 10% (Eurobio, Courtaboeuf, France) 4 h after drug administration. Sera from animals were collected over time and IgG titers were determined by serial dilution on SRBC by flow cytometry using a fluorescent anti human IgG Ab (Dako, Glostrup, Denmark).

**Flow cytometry.** Baboon leukocytes were purified by Ficoll density gradient separation from whole blood following red blood cell lysis. Fluorescent mAbs against human CD3 (SP34-2), CD4 (L200), CD8 (RPA-T8), CD11b (ICRF44), CD16 (3G8), CD19 (HIB19), CD20 (2H7), CD25 (MA251), CD28 (CD28.2), CD45RA (5H9), CD69 (FN50), CD95 (DX2), CD127 (hIL-7R-M21), Foxp3 (236A/E7), Ki-67 (B56), HLA-DR (G46-6) and pSTAT5 (47/STAT5(pY694)) were purchased from BD Biosciences and all used at the dilution of 1/5. An anti-human CD127 (A019D5, dilution 1/10), and CCR7 (G043H7, dilution 1/5) were purchased from Biolegend. Anti-PD-1 (eBioJ105, dilution 1/10) was purchased from eBioscience, anti-human CD14 (TUK4, dilution 1/10) and anti-human CD33 (AC104.3E3, dilution 1/10) from Myltenyi and anti-human NKp46 (BAB281, dilution 1/10) from Beckman Coulter. Samples were acquired on a BD FACS LSRII flow cytometer (BD Biosciences) and analyzed with FlowJo software. The percentage of receptor occupancy (RO), was determined using flow cytometry staining with a non-competitive commercial anti-human CD127 mAb (hIL-7R-M21) and a competitive commercial anti-human CD127 mAb (A019D5). The percentage of free CD127+ cells in total CD127+ T cells was analyzed by flow cytometry.

**Elispot assays.** Baboon PBMCs were prepared from whole blood by Ficoll gradient centrifugation (GE Healthcare Life Science, Paris, France). Red blood cells were then lysed. Cells were harvested and washed twice at low speed to remove platelets. PBMCs were washed with media (RPMI 1640, Penistreptomycin, Glutamine, Non-essential amino acid, Pyruvic acid-Hepes, 10% of baboon sera) and added to PVDF plates (0.45 μm, Merck Millipore), which had been precoated with an IFN-γ antibody (non-human primate IFN-γ ELISPOT kit; R&D Systems, Minneapolis) according to the manufacturer's instructions. PBMCs were then stimulated with 40 UI tuberculin-purified protein derivative (PPD; Symbiotics Corporation, San Diego, CA, USA), IL-2 (600 UI/mL) or anti-human IL-7Rα antagonist mAb (10 μg/ml). In some conditions, CD25+ cells from PBMCs were depleted using a phycoerythrin (PE)-labelled anti-human CD25 (M-A251) plus anti-PE microbeads (Milteny) and a negative selection with autoMACS separator (Miltenyi Biotec). Incubation was performed overnight at 37 C°. Spots were evaluated using the ELISpot reader system (AID Ispot Spectrum) with the software version 7.0. Results were expressed as number of spots per $1 \times 10^5$ PBMC.

**In vitro human PBMC stimulation assay.** Human PBMCs, prepared as described above, were stained with a Cell Proliferation Dye eFluorTM 670 (CPD) (Thermofisher, 65-0840-85) according to manufacturer instructions. CPD-stained PBMCs were cultured in 96-well plate at $2 \times 10^6$ cells/mL in TexMACS medium (Miltenyi) supplemented with 100 U/mL penicillin and 100 mg/mL streptomycin. PBMCs were cultured when indicated with 5 ng/ml of human recombinant interleukine-7 (Biorad, PHP046), MHC-I restricted PepTivator Influenza A (H1N1) MP1 pool of peptides (Miltenyi, 130-097-285) at 0.4 nmol/ml, MHC-II restricted CEFT peptides pool (JPT Peptide Technologie) at 1 nmol/ml, MHC-I restricted CEF (PepTivator CEF MHC Class I Plus, Miltenyi, 130-098-426) at 0.4 nmol/ml, or with DynabeadsTM human T-Activator CD3/CD28 (Thermofisher, 11131D). In some conditions, PBMC were also cultured with 10 μg/ml of a humanized anti-IL-7Ralpha monoclonal antibody. Cells were stained with FITC-Annexin-V (BD Bioscience, 556547) and propidium iodide (PI) (Sigma Aldrich,

P4170-25MG) at the end of the culture. The percentage of proliferating cells (% of CPDlow cells) and viable cells (% of Annexin-V⁻ PI⁻ cells) was analyzed by flow cytometry (LSR II, BD Bioscience).

**Ex vivo baboon PBMC polyclonal activation assay.** Baboon PBMCs extracted at different time-points were cultured in 96-well plates at $2 \times 10^6$ cells/mL in Tex-MACS medium (Miltenyi) supplemented with 100 U/mL penicillin and 100 mg/mL streptomycin for 48 h. PBMCs were stimulated with coated anti-CD3 Ab (SP34; 1 µg/mL), human IL-2 (300 UI/mL) or human IL-7 (10 ng/mL; Miltenyi). T lymphocyte activation was measured by CD25 (MA251) and CD69 (FN50) staining, followed by flow cytometry.

**Ex vivo baboon PBMC metabolism evaluation.** Baboon T cells were purified by negative selection with a Pan T cell isolation kit (Miltenyi) and cultured in TexMACS medium at 37 °C/5% $CO_2$. T cells were analyzed with or without over-night activation in 24-well plates ($5 \times 10^5$ T cells/well) with a T cell Activation/Expansion kit from Miltenyi that consists of AntiBiotin MACSiBead™ Particles and biotinylated antibodies against non-human primate CD3 and human CD2 and CD28. The extracellular acidification rate (ECAR), which approximates glycolytic activity, and the mitochondrial oxygen consumption rate (OCR), which is a key metric of mitochondrial function, were measured using a Seahorse XF24 analyzer. Assays were performed in Seahorse XF-base medium supplemented with 10 mM glucose (Sigma), 2 mM glutamine (Life Technologies) and 1 mM pyruvate (Life Technologies). Mitochondrial stress assays were performed by successively adding oligomycin (1.5 µM; Sigma), CCCP (1 µM; Sigma) and Antimycin A + Rotenone (1 µM each; Sigma). In parallel, purified T cells were cultured over night with or without plate bound anti-CD3, IL-7 (10 ng/mL, Miltenyi) or with the Activation/Expansion kit (Miltenyi). To measure glucose uptake, cells were washed and cultured for 2 h in glucose-free medium containing 20 mM 2-NBDG (Invitrogen) prior to analysis by flow cytometry. 15 µl of each cell suspension ($6 \times 10^4$ cells) were used for luminometric ATP measurement using an Apo Biovision kit according to manufacturer's instructions (Cliniciences). Luminometry was measured with a Spark reader (TECAN).

**Immunohistochemistry.** Frozen sections (10 µm) were prepared from surgical skin biopsies. Slides were air-dried at room temperature overnight before acetone fixation for 10 min at room temperature. Sections were saturated with PBS containing 10% baboon serum, 2% normal goat and donkey serum, and 4% BSA. Sections were incubated overnight with primary Abs at 4 °C, followed by fluorescent secondary Abs as previously described[45,69]. T cell infiltration analysis was performed using rabbit anti-human CD3 Ab (Dako), and macrophage infiltration using a mouse anti-human CD68 Ab (clone KP1; Dako). Slides were analyzed with the AxioVision imaging software (Carl Zeiss, Le Pecq, France).

**Statistical analysis.** Continuous variables were expressed as the mean ± SEM, unless otherwise indicated, and compared with the Mann–Whitney non-parametric two-sided test. When multiple groups were compared, data were compared using one-way ANOVA with Dunn's test multiple comparisons. P values of < 0.05 were considered as statistically significant. All statistical analyses were performed on GraphPad Software (GraphPad Software, San Diego, CA).

## Data availability

The dataset generated during the current study are available in the GEO repository, GEO103643, and is available from the corresponding author on reasonable request. All data generated or analysed during this study are included in this published article (and its supplementary information files).

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

## Acknowledgements

This work was supported by the French Public Bank of Investment (Effimab grant n°I 1302011 W), the Fondation pour la Recherche Médicale (grant number LBS20130627235) and the Région of Pays de la Loire. This work was realized in the context of the IHU-Cesti project, which received French government financial support managed by the Agence Nationale de la Recherche via the "Investment Into The Future" program ANR-10-IBHU-005. The IHU- Cesti project is also supported by Nantes Metropole and the Pays de la Loire Region. This work was also supported by the FP7 VISICORT project that has received funding from the European Union's Seventh Framework Programme for research, technological development and demonstration under grant agreement 602470. This work was realized in the context of the LabEX IGO program supported by the National Research Agency via the investment of the future program ANR-11-LABX-0016-01. We thank Dr. Claire Pecqueur-Hellman for her assistance in the use of Seahorse XF24 Analyzer. We thank Dr. Aurore Morello for her kind review of the manuscript.

## Author contributions

Conceived the study: B.V., N.P. Designed and supervised the experiments: L.B., C.M., N.D., B.V., N.P. Performed the experiments: L.B., C.M., L.J., H.L.M., J.H., D.M., V.T., V.D., S.P., E.L., E.N., B.M., S.L.B. Analyzed and interpreted data: L.B., C.M., L.J., H.L.M., A.D., R.D., S.B., J.P.S., G.B., N.D., N.P. Wrote the paper: L.B., B.V., N.P.

## Additional information

**Competing interests:** The authors of this manuscript have conflicts of interest to disclose: CM, JPS, SB, BV and NP are shareholders of OSE Immunotherapeutics, a company owning patented anti-IL-7 receptor antagonists in clinical development. The remaining authors declare no competing interests.

