## [Peer Review File · Nature Communications]

Reviewers' comments:

Reviewer #1 (IL7, T homeostasis)(Remarks to the Author):

The manuscript by Belarif et al presents clear evidence that, while not leading to broad lymphoid depletion (in contrast to what happens in mouse models) anti-IL-7R α antibodies can effectively prevent antigen-specific chronic inflammation in non-human primates. In addition, the manuscript points out that anti-IL-7R α antibodies may have not only antagonistic but also, depending on their target epitopes, agonistic effects. This appears to determine their ability to prevent antigen-specific inflammation, with "dual effect" mAbs being non-effective.

MAJOR COMMENTS

1. Details of how the antibodies were generated are briefly mentioned in the results. They should be included, with sufficient information, in the Methods.
2. Not clear why site 1/2b antibodies do not block TSLP-induced CCL17 production (sup fig 1) if TSLPR heterodimerization involves site 2b (as mentioned in the Discussion, page 15). Please discuss.
3. Isotypic controls should be shown in at least some of the in vitro experiments. For instance, it is essential to show these controls for all the data presented in supplementary figure 1. Also, show isotypic controls for Figure 3C/D.
4. Also regarding Supp fig 1, since in upper panel A 10ug/ml dose is shown as being representative, the titration data below should also include this concentration.
5. In fact, it seems obvious from both upper panel A and lower panel A in Supp fig 1, that S1/2b antibodies, although having a strong effect, are slightly less efficient in inhibiting IL-7-dependent STAT5 phosphorylation than the site 1 Ab, not being able to completely abrogate p-STAT5 (contrary to site 1). A small fraction of p-STAT5 positive cells may still have an important biological impact. This appears to be paradoxical. However, it is curiously coincident with data from transcriptome analysis, which in fact indicates (fig. 4B) that site 1/2b Ab is not as efficient as the site 1 Abs in reversing cluster 3 genes (highly upregulated by IL-7). Could this, rather than (or in addition to) the induction of MEK/ERK pathway by the site 1 Abs be the reason for the different ability of Abs to prevent inflammation in vivo? i.e. could it be that SOME IL-7 effects (preserved by the site 1/2b Abs) are actually anti-inflammatory? For example IKZF4, which is highly induced by IL-7 and not clearly affected by site 1/2b Ab, was reported to prevent TH17 polarization (Liu et al, JBC 289(18):12446-56).
Could the authors experimentally exclude this hypothesis?
5. In figure 2B, site 1/2b mAb images refer to IgG1 or IgG4? Indicate and show similar images for the other isotype (either in main fig or as supp data).
6. Given the functional data presented in Fig 2, the data in figure 3 should include, as a minimum, the same Abs and thus data for Site 1/2b IgG1 should also be included in fig. 3 (to strengthen the idea that differences in fig 2 are in fact related to site 1 x site 1/2b antagonistic x dual effects).
7. Similarly, why such considerable differences between the two site 1 Abs in terms of gene expression profiles induced in human PBMCs (fig. 4A)? How about site 1/2b Abs? Would they also give such heterogeneous results. This should be shown.

8. Check whether differential effects of Abs on IL-7-mediated signaling occur, similar to human PBMCs, in baboon PBMCs.

9. Supp Fig 5: internalization of CD127 or transcriptional downregulation? Given the time frame of the analyses both are possible. To properly address this and argue that the antibodies induce internalization (as stated in page 16 of Discussion): 1) check transcript levels of IL7R at same time points; 2) perform analysis of surface x total IL-7Ra levels.

MINOR

1. In the introduction (page 3) “the specific IL-7Ra” is not accurate. As the authors know (address elsewhere in the manuscript) this subunit is shared with the TSLP receptor. This should be corrected.

2. Reorganize the order of supplementary figures to follow more logically the order in which they are first mentioned in the main text (e.g. Supp fig 5 should be Supp fig 2).

Reviewer #2 (Tolerance/anergy, T activation)(Remarks to the Author):

In this study, Belarif et al generated and studied antibodies against the IL-7 receptor. They found that all IL-7 blocking anti-IL7Ra mAbs prevented JAK/STAT signaling but, depending on the epitope, some mAbs also presented agonist properties that modified the transcriptome of peripheral blood mononuclear cells. The IL-7 blocking mAbs with dual agonist and antagonist properties lacked efficacy in vivo. In contrast, an “antagonist-only” anti-IL7Ra mAb induced long-term control of skin inflammation in non-human primates mediated by DTH response to PPD after prior BCG vaccination without affecting polyclonal T cell responses to non-specific stimulus such as anti-CD3/CD28 mAbs. However, strong re-challenge with specific antigen by re-vaccination with BCG restored responses to tuberculin. Based on these findings the authors conclude that chronic antigen-specific memory T cell responses can be controlled by anti-IL7Ra mAbs that have full antagonist properties and such antibodies can be useful for promoting and maintaining remission in T-cell mediated chronic inflammatory diseases.

The study is interesting and informative and such anti-IL-7 receptor antibodies might have direct therapeutic applications. Several points require further clarification before conclusions can be made:

1. A key concern regarding the potential therapeutic efficacy of these antibodies in vivo stems from the observation that re-vaccination restored T cell responses to specific antigen. This result suggests that exposure to increased density of self-antigen(s) which are the drivers of T cell activation in chronic inflammatory diseases might reverse the ability of IL7Ra antibodies to induce hypo-responsiveness in vivo. This could also occur when inflammatory conditions drive the upregulation of costimulatory molecules which enhance TCR-mediated signaling. The authors should clarify this point in their manuscript and discuss the potential clinical implications.

2. To assess specificity of the antibodies, in supplementary Figure 1B, the investigators tested whether these antibodies inhibited TSLP-induced CCL17 secretion by dendritic cells. It is necessary to provide one sentence justification and rationale for this approach at this point of the manuscript as they did in

the discussion.

3. Supplementary Figure 5A: What is the mechanism of the decreased MFI for CD127 in the recipients of site-1 IgG4 mAb?

4. The authors stated that analysis of T cell subsets did not reveal any significant modification of naïve, effector or central memory subsets within CD4⁺ or CD8⁺ T cell populations, in Treg cells or in exhausted PD1⁺ CD8⁺ T cells. None of the Treg and T-exhausted cell data are shown. These data are significant and should be included in a supplemental figure. For identification of exhausted CD8⁺ T cells, expression of PD-1 alone is not appropriate because PD-1 is an activation marker. Instead, assessment of a combination of exhaustion markers such as PD-1, CTLA4, LAG3, TIGIT and TIM3, should be done.

5. Figures 5 and 6 show that in animals treated with site-1/2b anti-IL-7R α mAb, tuberculin-induced skin inflammation was visible again after BCG re-vaccination in parallel with recovered responsiveness to tuberculin. These findings indicate that the blocking effects of site-1/2b anti-IL-7R α mAb are transient and after potent restimulation to the specific antigen responsiveness is restored. What is the mechanism of the transient effect of these antibodies in suppressing antigen-specific memory responses?

6. Figure 7: As indicated in the figure legend, T cell activation was assessed after ex-vivo culture (48 hours). What is the time (days) indicated in the x axis in panels A and B? I guess these must refer to the time after immunization. In this case, these timepoint indications should be kept the same as in panels C and D (which show time in weeks) to avoid confusion.

7. While refereeing to the results shown in figure 7B, the authors state "No significant difference was observed after treatment compared to pretreatment activation status" (page 12). The expression of CD69 and CD25 should be shown in cells isolated ex vivo and prior to in vitro treatment (first part of the panels in Figure 7B).

8. Page 13: Based on their findings, the authors concluded that antagonist anti-IL7R α mAbs do not provoke general immunosuppression but induce long-term immune tolerance in non-human primates through clonal deletion or intrinsic control of antigen-specific memory T cells after antigen rechallenge. However, the data show that after re-immunization with BCG, there is expansion of antigen-specific cells in animals that were previously treated with site-1/2b antibodies. This finding indicates that antigen-specific cells were not deleted but remained present and capable of responding to the specific antigen after more potent stimulation such as re-vaccination with BCG followed by PPD.

Reviewer #1:

The manuscript by Belarif et al presents clear evidence that, while not leading to broad lymphoid depletion (in contrast to what happens in mouse models) anti-IL-7R α antibodies can effectively prevent antigen-specific chronic inflammation in non-human primates. In addition, the manuscript points out that anti-IL-7R α antibodies may have not only antagonistic but also, depending on their target epitopes, agonistic effects. This appears to determine their ability to prevent antigen-specific inflammation, with “dual effect” mAbs being non-effective.

MAJOR COMMENTS:

1. *Details of how the antibodies were generated are briefly mentioned in the results. They should be included, with sufficient information, in the Methods.*

OK, the following paragraph was added at the beginning of the methods section.

Anti-IL7Ra mAbs:

Anti-human IL7Ra mAbs were generated by CDR-grafting technology after sequencing of hybridoma generated from Lou/C rats immunized with recombinant human IL-7Ra protein. CDR were grafted in most appropriate human VH/VL frameworks and combined with a human Fc domain from IgG4 (S228P mutation preventing Fab-arm exchange) or IgG1 (E333A mutation increasing Fc γ R affinity and cytotoxic functions) isotypes. Resulting humanized antibodies were then recombinantly expressed in HEK cells. The site-1 anti-IL7Ra mAb clone 1A11, of the human IgG1 isotype, was identified in the literature (WO/2011/09259) and has been similarly recombinantly expressed in HEK cells. Antibodies have been purified from cell supernatant by a capture chromatography on Protein-A-Sepharose. Purity by HPLC was > 90%.

2. *Not clear why site 1/2b antibodies do not block TSLP-induced CCL17 production (sup fig 1) if TSLPR heterodimerization involves site 2b (as mentioned in the Discussion, page 15). Please discuss.*

The recent manuscript by Verstraete K et al. (Nature Communication 2017), describing the interaction between IL7Ra and TSLPR, showed that single mutation in TSLPR site-2b (named site III in their manuscript) had no apparent effect on TSLP signaling while combinations of mutations reduced (but did not abrogate) activation of STAT5 in response to TSLP (Figure 4b of Verstraete’s manuscript). The conclusion is that TSLPR site-2b is involved in efficient signaling. Furthermore, the IL7Ra residues involved in the interaction with TSLPR and those described previously to be predicted for the interaction of IL7Ra with the gamma-chain (Walsh Immunological review 2012; McElroy et al Structure 2009) are structurally close to each other but not identical (only 2 out of 7 identic residues). Finally, while Verstraete et al. described similar structural interaction of TSLP or IL7 with IL7Ra at site-1, when comparing residues in IL7Ra site-1 involved in the interaction with TSLP (in this Verstraete’s manuscript) with those involved in the interaction with IL7 (McElroy et al. Structure 2009), we can also observe regional similarity, but not exactly same residues involved in direct contact. None of the antibodies evaluated in our study, targeting at site-1 or site-1/2b, displayed efficient (>20%) inhibition of TSLP-induced TARC secretion (which is also mediated by pSTAT5) as opposed to an antagonistic anti-TSLPR antibody. This might reflect the importance of differences at site-1 and site-2b between TSLP and IL7 interaction with IL7Ra. The following sentences were added in the discussion section:

The heterodimerization of this site 2b with TSLPR has also been recently confirmed and demonstrated to play a role in receptor signaling, as already predicted for IL-7R α and the γ -chain⁶³. However, while

IL7R α employs similar, but not identical, set of residues at site-1 to interact with IL-7 and TSLP⁶³, neither site-1 nor site1/2b anti-IL7R α mAbs displayed significant inhibition of TSLP-induced TARC secretion by human dendritic cells.

3. *Isotypic controls should be shown in at least some of the in vitro experiments. For instance, it is essential to show these controls for all the data presented in supplementary figure 1. Also, show isotypic controls for Figure 3C/D.*

We have added Ig control in supplementary Figure 1 A and B (new figure below). Regarding supplementary Figure 1C, ADCC was observed with an IgG1 anti-IL7Ra mAb and not with 2 different IgG4 isotype, according to the high versus low affinity of IgG1 versus IgG4 for Fc γ R. To further confirm the specific ADCC effect of the IgG1 anti-IL7R mAb we have added new data on supplementary Figure 1Cright showing this IgG1 antibody induced ADCC on human T-cell line expressing high level of IL-7R (DND41 cell line: T-cell Acute Lymphoblastic Leukemia) and not against another human T-cell line expressing very low level of IL-7R (Jurkat cell line: T-cell Acute Lymphoblastic Leukemia).

Supplementary figure 1

To answer to the point related to isotypic controls in Figure 3C/D, as well as question n°6 on signaling data with site-1/2b IgG1 antibody and question n°8 on baboon signaling data, we have generated a novel supplementary Figure 6 showing new and independent experiments confirming the agonist effect of site-1 anti-IL7Ra antibody of either IgG1 or IgG4 isotype on Pi3K and ERK pathway as opposed to IgG1 or IgG4 site-1/2b antibodies and IgG1 or IgG4 isotypic controls on human PBMC (A) and baboon PBMC (B). The figure below shows that IgG1 and IgG4 isotype controls do not induce pPi3K in contrast to site-1 IgG1 or IgG4 mAbs. Comparable results were observed on pERK while IgG1 isotype induced some degree of pERK in comparison to IgG4, although less than site-1 IgG1 mAbs.

Supplementary figure 6

A

B

4. Also regarding Supp fig 1, since in upper panel A 10ug/ml dose is shown as being representative, the titration data below should also include this concentration.

This has been added to the Supp Fig1 (A right).

In fact, it seems obvious from both upper panel A and lower panel A in Supp fig 1, that S1/2b antibodies, although having a strong effect, are slightly less efficient in inhibiting IL-7-dependent STAT5 phosphorylation than the site 1 Ab, not being able to completely abrogate p-STAT5 (contrary to site 1). A small fraction of p-STAT5 positive cells may still have an important biological impact. This appears to be paradoxical. However, it is curiously coincident with data from transcriptome analysis, which in fact indicates (fig. 4B) that site 1/2b Ab is not as efficient as the site 1 Abs in reversing cluster 3 genes (highly upregulated by IL-7).

We agree that some very small difference could be found between all these antibodies in term of pSTAT5 inhibition potency. This might be explained also by small but potentially relevant differences in affinity. To our experience, there is no significant and reproducible difference between site-1 and site-1/2b antibodies potency for pSTAT5 inhibition. The novel figure with new data is probably less ambiguous for this point. The most reproducible, even small, difference found in nearly all our experiment is a more favorable IC50 for IgG1 anti-IL7Ra mAbs as compared to same antibody in IgG4 format, in both site-1 or site-1/2b mAbs. We agree that based on RNA-SEQ signature, the site-1/2b antibody seems less, although not significantly, efficient than the two site-1 antibodies compared in parallel. We have no explanation for this difference, despite a different epitope which might be relatively less efficient at blocking IL7 signaling but on the contrary not agonist on PI3K and ERK pathways and associated with preclinical efficacy in a memory T-cell induced inflammation model in non-human primates.

The following sentences in the result section have been modified: Interestingly, all mAbs exhibited similar effects by preventing cluster-1 and cluster-3 modification (somewhat lower for the site-1/2b than site-1 mAbs on cluster 3) but without impact on the mixed cluster-2 (Figure 4B, bottom).

Could this, rather than (or in addition to) the induction of MEK/ERK pathway by the site 1 Abs be the reason for the different ability of Abs to prevent inflammation *in vivo*? i.e. could it be that SOME IL-7 effects (preserved by the site 1/2b Abs) are actually anti-inflammatory?

To our knowledge and based on the existent literature, there is no evidence that IL7 might exercise some anti-inflammatory effect on effector T cells. Regarding anti-inflammatory Tregs cells, it was recently described that abrogation of IL7 or IL7R signaling impairs Tregs suppressive capacity and Treg-mediated allograft tolerance (Schmaler M. et al PNAS 2015), suggesting that IL-7 in mice might indirectly contributes to the anti-inflammatory activity of Tregs. However, this report is in contradiction to previous reports in human describing deleterious effect of IL7 signaling on Treg suppressive functions (Heninger AK et al JI 2012; Van Amerlsfort JM et al Arthritis Rheum 2007; Ruprecht CR et al J Exp Med 2005; Allgäuer A et al JI 2015).

For example IKZF4, which is highly induced by IL-7 and not clearly affected by site 1/2b Ab, was reported to prevent TH17 polarization (Liu et al, JBC 289(18):12446-56). Could the authors experimentally exclude this hypothesis?

We cultured human PBMC from 3 different donors for 3 hours (as performed during RNA-SEQ experiment) with 10 μ g/ml of two mAbs targeting at the site-1 (IgG4 or IgG1), the mAb targeting at site-1/2b (IgG4) and compared to medium and isotypic IgG4 control the IKZF4 mRNA expression by qRT-PCR. The results below showed that none of the antibody, targeting at site-1 or site-1/2b, increased or decreased significantly mRNA expression of IKZF4. This novel experiment confirms previous RNA-SEQ data on supplementary table 1: IKZF4 gene is not present among the 481 genes differentially and significantly expressed in human PBMC incubated with anti-human site-1 or site-1/2b mAbs as compared to control condition.

5. In figure 2B, site 1/2b mAb images refer to IgG1 or IgG4? Indicate and show similar images for the other isotype (either in main fig or as supp data).

The previous figure showed skin IHC data from animals treated with site-1/2b IgG4 mAb and site-1 IgG4 mAb. We have updated the figure with IHC data from the three groups of animals as suggested: site-1/2b IgG1, site-1/2b IgG4, and site-1 IgG4 mAb. Figure below.

6. Given the functional data presented in Fig 2, the data in figure 3 should include, as a minimum, the same Abs and thus data for Site 1/2b IgG1 should also be included in fig. 3 (to strengthen the idea that differences in fig 2 are in fact related to site 1 x site 1/2b antagonistic x dual effects).

As explain above, to answer to the point relating to isotypic controls in Figure 3C/D, as well as question n°6 on signaling data with site-1/2b IgG1 antibody and question n°8 on baboon signaling data, we have generated a novel supplementary Figure 6 showing new and independent experiments confirming the agonist effect of site-1 anti-IL7Ra antibody of either IgG1 or IgG4 isotype on Pi3K and ERK pathway as opposed to IgG1 or IgG4 site-1/2b antibodies and IgG1 or IgG4 isotypic controls on human PBMC (A) and baboon PBMC (B). The figure below shows that indeed the site-1/2b IgG1 mAb does not induce for example pPI3K in contrast to site-1 IgG4 or IgG1 mAbs. Comparable results were observed on pERK while IgG1 isotype induce some degree of pERK in comparison to IgG4, although less than site-1 IgG1 mAbs.

Supplementary figure 6

7. Similarly, why such considerable differences between the two site 1 Abs in terms of gene expression profiles induced in human PBMCs (fig. 4A)? How about site 1/2b Abs? Would they also give such heterogeneous results. This should be shown.

The reviewer is right that transcriptomic modifications induced by mAbs and measured by RNA-SEQ seems quite heterogeneous on Venn diagram (Figure 4A top). The aim of this figure was to show that, despite a common gene signature between site-1 and site-1/2b mAbs of 61 genes, site-1/2b mAb induced very weak mRNA modification in accordance to its non-agonist property measured on signaling experiments. In contrast, the 2 site-1 mAbs, which induced PI3K and ERK pathways, clearly induced important and significant transcriptomic modification when applied on human PBMC. To extract significant and relevant mRNA modification from the RNA-SEQ analysis we used conventional threshold of p-value < 5% and fold-change > 1.5. Based on these threshold, nearly 47% of mRNA modification is shared between the 2 site-1 mAbs (61+78=139 common genes on a total of 292 or 297 genes for each site-1 mAb). As shown on the example on Figure 4A bottom, some genes reach both thresholds such as SIPA1L3, OX40L or PLK3. However, while other genes appeared upregulated with the 2 site-1 mAb and not with the site-1/2b mAb, they reach significance only for one of the 2 site-1 mAb (for example: XCL2, PIM3, CD6, KCNA3, KDM6B). Due to important inter-individual variability in human samples for RNA-SEQ analysis, the representation of significant gene modification with Venn diagram probably shows more heterogeneous response between the 2 site-1 mAbs than it is really the case. Indeed, analysis of gene pathways enriched in the signature of each site-1 mAb showed shared upregulated pathways with the two site-1 mAb (Supplementary Figure 7).

8. Check whether differential effects of Abs on IL-7-mediated signaling occur, similar to human PBMCs, in baboon PBMCs.

As explain above, to answer to the point relating to isotypic controls in Figure 3C/D, as well as question n°6 on signaling data with site-1/2b IgG1 antibody and question n°8 on baboon signaling data, we have generated a novel supplementary Figure 6 showing new and independent experiments confirming the agonist effect of site-1 anti-IL7Ra antibody of either IgG1 or IgG4 isotype on Pi3K and ERK pathway as opposed to IgG1 or IgG4 site-1/2b antibodies and IgG1 or IgG4 isotypic controls on human PBMC (A) and baboon PBMC (B). The figure below shows that site-1 mAbs also induce pPI3K and pERK in baboons as previously described on human cells.

Supplementary figure 6

9. Supp Fig 5: internalization of CD127 or transcriptional downregulation? Given the time frame of the analyses both are possible. To properly address this and argue that the antibodies induce internalization (as stated in page 16 of Discussion): 1) check transcript levels of IL7R at same time points; 2) perform analysis of surface x total IL-7Ra levels.

The reviewer is right, both internalization and mRNA downregulation mechanism could be possible in vivo. We interpreted our results as mainly mediated through internalization since we found in vitro (figure below) that our antibody targeting at site-1 or antibodies from others targeting also at site-1 induce receptor internalization while it was not the case with our site-1/2b antibody. This in-vitro observation correlated with our in-vivo observation in baboons (MFI decreased only with site-1 Ab) and data from other (Pfizer group) reporting in an abstract (*International Cytokine Society 2013 Abstract 144 by Kern B et al.: Receptor occupancy an internalization of an anti-IL7 receptor antibody*) that their anti-IL7Ra mAb (known to target at site-1 based on their patent epitope disclosure, Ab site-1 #3 in our figure below) induced ex-vivo significant internalization in monkey and human. This was confirmed later when they published these data in monkey (Kern B et al. Cytometry B Clin Cytom 2016). To exclude the possibility that anti-IL7Ra mAbs also induced direct mRNA downregulation, we cultured human PBMC with site-1 and site-1/2b mAbs and analyzed CD127 mRNA expression after 3 hours of incubation at 37°C. The data below shows that anti-IL7Ra mAbs did not induce CD127 mRNA decrease in vitro as previously observed in our RNA-SEQ analysis on supplementary table 1: CD127 gene is not present among the 481 genes differentially and significantly expressed in human PBMC incubated with anti-human site-1 or site-1/2b mAbs as compare to control condition.

Finally, since we cannot exclude one mechanism from another in-vivo, we modified the discussion sentence according to the point mentioned by the reviewer: In line with our site-1 specific mAb in vivo results in baboons (Supplementary Figure 2A), the same group described in a different report that their anti-IL7Ra mAb was able to induce IL7Ra internalization/downregulation in cynomolgus monkeys⁵².

In-vitro internalization assay on human PBMC incubated 30 min with 1 µg/ml of PH-rodo conjugated anti-IL7Ra mAbs

CD127 mRNA expression in human PBMC incubated 3 hours with anti-IL7Ra mAbs.

MINOR COMMENTS:

1. In the introduction (page 3) “the specific IL-7R α ” is not accurate. As the authors know (address elsewhere in the manuscript) this subunit is shared with the TSLP receptor. This should be corrected.

OK, this was corrected. The word “specific” was deleted

2. Reorganize the order of supplementary figures to follow more logically the order in which they are first mentioned in the main text (e.g. Supp fig 5 should be Supp fig 2).

OK this was reorganized and corrected

Reviewer #2:

In this study, Belarif et al generated and studied antibodies against the IL-7 receptor. They found that all IL-7 blocking anti-IL7R α mAbs prevented JAK/STAT signaling but, depending on the epitope, some mAbs also presented agonist properties that modified the transcriptome of peripheral blood mononuclear cells. The IL-7 blocking mAbs with dual agonist and antagonist properties lacked efficacy in vivo. In contrast, an “antagonist-only” anti-IL7R α mAb induced long-term control of skin inflammation in non-human primates mediated by DTH response to PPD after prior BCG vaccination without affecting polyclonal T cell responses to non-specific stimulus such as anti-CD3/CD28 mAbs. However, strong re-challenge with specific antigen by re-vaccination with BCG restored responses to tuberculin. Based on these findings the authors conclude that chronic antigen-specific memory T cell responses can be controlled by anti-IL7R α mAbs that have full antagonist properties and such antibodies can be useful for promoting and maintaining remission in T-cell mediated chronic inflammatory diseases. The study is interesting and informative and such anti-IL-7 receptor antibodies might have direct therapeutic applications. Several points require further clarification before conclusions can be made:

1. *A key concern regarding the potential therapeutic efficacy of these antibodies in vivo stems from the observation that re-vaccination restored T cell responses to specific antigen. This result suggests that exposure to increased density of self-antigen(s) which are the drivers of T cell activation in chronic inflammatory diseases might reverse the ability of IL7R α antibodies to induce hyporesponsiveness in vivo. This could also occur when inflammatory conditions drive the upregulation of costimulatory molecules which enhance TCR-mediated signaling. The authors should clarify this point in their manuscript and discuss the potential clinical implications.*

The reviewer is right, while in mice anti-IL7R α mAbs have been described in some models to induce long-term tolerance to allograft or in type 1 diabetes, our report is the first to describe this long-term effect in larger species. In mice, breaking tolerance has not been tried so far to our knowledge, using strong vaccine strategy with live-attenuated bacteria expressing the antigen (*i.d.* tuberculin) but also multiple danger-signals. When we re-challenged monthly monkeys with immunogenic intra-dermal injections of the isolated antigen (*i.d.* tuberculin protein), animals were non-responders demonstrating the durable and robust long-term hyporesponsiveness. Our novel (discussed below) experimental data showed that anti-IL7R α antagonistic mAb strongly impacted survival of antigen-specific memory T cells after rechallenge in vitro. This mechanism of action explains the long-term action observed *in-vivo* in monkeys where we also found a selective disappearance of memory T cells capable to respond to the tuberculin antigen. This clonal deletion mechanisms highlighted in our study in monkey is also supported by a recent report in autoimmune neuroinflammation (EAE) mouse model which described that anti-IL7R α treatment inhibited the expansion of autoantigen-specific T cells by inducing apoptosis of recently activated autoreactive T cells and leading to strong but selective depletion of antigen-specific memory T cells (Lawson et al. Clin Immunol 2015).

While most rodent models using anti-IL7R α mAbs reported the induction of regulatory T cells, here we did not observe such effect. Immune tolerance supported by regulatory T cells might be more robust since capable of inhibiting also novel important immune re-challenge. In contrast, immune tolerance mediated by selective memory T-cell clonal deletion does not prevent re-immunisation coming from the priming of novel naive T cells then capable to re-induce immune responses. However, in terms of safety profile, it is comfortable to know that we have the possibility to reverse the induced immune tolerance if it developed against undesirable antigen (*e.g.* pathogenic viruses). Altogether, this might also suggest that while anti-IL7R α demonstrated the potential for a long-term anti-inflammatory effect in larger species, it would probably require iterative injections in patients in case of relapse.

This point is now more clearly discussed: Finally, we found that animals re-vaccinated long after drug withdrawal recovered a normal immune response, in accordance with the mechanism of action of antigen-specific memory T cells clonal deletion upon treatment which should not affect de-novo priming of naïve T cells particularly after drug elimination. Hence, while anti-IL-7R α mAb treatment demonstrated preclinical efficacy, preventing relapse in patients suffering from chronic inflammatory and/or autoimmune diseases will probably require iterative administrations. Furthermore, while therapeutic strategies inducing robust immune tolerance might suffer from their safety profile (becoming tolerant to an unwanted pathogen), here we showed that re-vaccination with live-attenuated pathogen was sufficient to generate de novo antigen-specific immune responses.

2. *To assess specificity of the antibodies, in supplementary Figure 1B, the investigators tested whether these antibodies inhibited TSLP-induced CCL17 secretion by dendritic cells. It is necessary to provide one sentence justification and rationale for this approach at this point of the manuscript as they did in the discussion.*

OK, the following sentence was added in the result section: IL-7R α interacts also with the TSLPR chain to form the heterodimeric receptor of the TSLP cytokine

3. *Supplementary Figure 5A: What is the mechanism of the decreased MFI for CD127 in the recipients of site-1 IgG4 mAb?*

Receptor internalization and/or mRNA downregulation mechanism could explain the decreased in CD127 MFI in vivo. As discussed in the manuscript, we interpreted our results as mainly mediated through internalization since we found in vitro (figure below) that our antibody targeting at site-1 or antibodies from others targeting also at site-1 induce receptor internalization while it was not the case with our site-1/2b antibody. This in-vitro observation correlated with our in-vivo observation in baboons (MFI decreased only with site-1 Ab) and data from other (Pfizer group) reporting in an abstract (*International Cytokine Society 2013 Abstract 144 by Kern B et al.: Receptor occupancy an internalization of an anti-IL7 receptor antibody*) that their anti-IL7R α mAb (known to target at site-1 based on their patent epitope disclosure, Ab site-1 #3 in our figure below) induced ex-vivo significant internalization in monkey and human. This was confirmed later when they published these data in monkey (Kern B et al. *Cytometry B Clin Cytom* 2016). To exclude the possibility that anti-IL7R α mAbs also induce direct mRNA downregulation, we cultured human PBMC with site-1 and site-1/2b mAbs and analyzed CD127 mRNA expression after 3 hours of incubation at 37°C. The data below shows that anti-IL7R α mAbs do not induce CD127 mRNA decrease in vitro as previously observed in our RNA-SEQ analysis on supplementary table 1: CD127 gene is not present among the 481 significant genes differentially expressed in human PBMC incubated with anti-human site-1 or site-1/2b mAbs as compare to control condition.

Finally, since we cannot exclude one mechanism from another in-vivo, we modified the discussion sentence according to the point mentioned by the reviewer: In line with our site-1 specific mAb in vivo results in baboons (Supplementary Figure 2A), the same group described in a different report that their anti-IL7R α mAb was able to induce IL7R α internalization/downregulation in cynomolgus monkeys⁵².

In-vitro internalization assay on human PBMC incubated 30 min with 1 µg/ml of PH-rodo conjugated anti-IL7Ra mAbs

CD127 mRNA expression in human PBMC incubated 3 hours with anti-IL7Ra mAbs.

4. The authors stated that analysis of T cell subsets did not reveal any significant modification of naïve, effector or central memory subsets within CD4+ or CD8+ T cell populations, in Treg cells or in exhausted PD1+ CD8+ T cells. None of the Treg and T-exhausted cell data are shown. These data are significant and should be included in a supplemental figure. For identification of exhausted CD8+ T cells, expression of PD-1 alone is not appropriate because PD-1 is an activation marker. Instead, assessment of a combination of exhaustion markers such as PD-1, CTLA4, LAG3, TIGIT and TIM3, should be done.

Indeed, the data of complete immunophenotyping (including Tregs and PD1+ CD8 T cells) were already presented in a Supplementary Figure 4, however the mention to the supp Fig in the text was missing. This is now corrected. The reviewer is right that PD-1 is not a specific marker of exhaustion and could be also express by T cells just after activation. However, due to the difficulty to find commercial fluorescent antibodies cross-reacting with proteins of baboons, we did not evaluate CTLA4, LAG3, Tim3 or TIGIT expression which would have improved exhaustion marker analysis. Here, we did not observe modification of the expression of PD-1 over-timer in CD8 T cells. This means that CD8 T cell activation and/or CD8 T cell exhaustion does not increase after treatment. This point has been clarified in the manuscript: No significant increase in natural CD4+ regulatory T cells (Tregs) or activated/exhausted CD8+ T cells (based on PD-1 expression) was observed (Supplementary Figure 4).

5. *Figures 5 and 6 show that in animals treated with site-1/2b anti-IL-7R α mAb, tuberculin-induced skin inflammation was visible again after BCG re-vaccination in parallel with recovered responsiveness to tuberculin. These findings indicate that the blocking effects of site-1/2b anti-IL-7R α mAb are transient and after potent restimulation to the specific antigen responsiveness is restored. What is the mechanism of the transient effect of these antibodies in suppressing antigen-specific memory responses?*

As discussed above, in most mouse studies where anti-IL7R α mAbs induce immune tolerance, authors reported a significant role of regulatory T cells. Only one mouse study (in neuroinflammation) reported that anti-IL7R α treatment inhibited the expansion of autoantigen-specific T cells by inducing apoptosis of recently activated autoreactive T cells and leading to strong but selective clonal deletion of antigen-specific memory T cells (Lawson et al. Clin Immunol 2015).

In our study, all our previous and novel data (exposed below) on the mechanism of action argue for the **selective elimination of antigen-specific memory T cells** after antigenic challenge (tuberculin challenge in vivo) and anti-IL-7R α mAb treatment, a phenomenon named **clonal deletion**:

- Monthly re-challenged baboons with immunogenic intra-dermal injections of the isolated tuberculin protein remained tolerant for a long-period of time (> 1 year) even after drug elimination. We found that mAb treatment did not induce decrease in peripheral T lymphocytes numbers, defect in polyclonal T-cell activation nor peripheral T-cell metabolic deficiencies (Figure 8 and Supp Fig 4). In contrast, we found a significant decreased ex-vivo of tuberculin-specific IFN γ secreting memory T cells by ELISPOT after treatment (Figure 5C). This decreased was maintained even after treatment withdrawal demonstrating it is also durable over time as observed for the in-vivo tolerance.
- The unresponsiveness *ex-vivo* remained observed even after elimination of CD25^{high} cells from PBMC suggesting that Tregs (which are not increased in vivo) are not in play in this inhibition *ex-vivo* (Figure 5D). Similarly, addition of high concentration of IL-2 in ELISPOT assays did not restore IFN- γ responses excluding an anergic state of antigen-specific memory T cells (Figure 5D).
- Interestingly, one animal did not present inhibition of skin inflammation and, in parallel, this animal did not display the decrease in IFN γ -secreting cells *ex vivo* after tuberculin restimulation (Supplementary Figure 9).
- Finally, after re-vaccination up to 1 year after treatment withdrawal, we found that animals recovered their skin inflammation in vivo and in parallel recovered also their frequency in IFN γ -secreting cells *ex vivo* in response to tuberculin stimulation (Figure 5C).

Altogether, the only factor correlated with in-vivo responsiveness is the frequency of antigen-specific memory cells (see Figure 5A vs 5C). Unfortunately, tetramers technology is not available in baboons to clearly demonstrate this point as it could be performed in mouse or human.

To demonstrate more specifically how the antibody induces clonal deletion after antigenic stimulation, we performed novel experiments on human cells. Indeed, while the biology of IL7 in sustaining survival and homeostasis of thymocytes and naïve T cells has been extensively reported in mouse, the role of the IL7 cytokine in human T cells biology has been less studied particularly the impact on human memory T cells antigenic restimulation. For example, homeostatic TCR signaling has been reported to enhance sensitivity to IL7, but some TCR signaling pathways have been also described to antagonize IL7 responsiveness (Carette F et al. Semin Immunol. 2012). It has also been reported that IL7 could increase the proliferation of human CD8⁺ memory T cells after viral peptides restimulation in-vitro (O'Connor et al. Immunology 2010) and that IL7 could promote the survival of human CD4⁺ effector/memory T cells by up-regulating anti-apoptotic Bcl-2 proteins through the JAK/STAT pathway (Chetoui N et al. Immunology 2010). Therefore, we first evaluated the impact of IL7 in-vitro on the proliferation and survival (using Annexin-V and Propidium Iodide staining) of human CD8⁺ and CD4⁺

memory T cells after MHC I-restricted (H1H1 flu) and MHC II-restricted (CMV, EBV, Flu, Tetanos) peptides restimulation respectively. We compared also the impact of IL7 on antigen-specific memory T cells restimulation with polyclonal stimulation using conventional anti-CD3+anti-CD28 mAbs stimulation. Interestingly, while adding IL7 had no significant impact on the proliferation of memory CD4 or CD8 T cells after 3 days of culture and only a modest (non-significant) positive effect on memory T cells survival (Figure 7A below), we found at late time-point (after 8 days of stimulation) that **IL-7 significantly and strongly improve memory CD4 and CD8 human T cells survival after antigen restimulation (Figure 7B)**. In contrast, no significant impact was found on polyclonal stimulation (anti-CD3/CD28) while cells proliferated vigorously and finally displayed very weak survival after 1 week of stimulation. Second, we performed similar experiment using this time IL7 alone or in combination with the anti-IL7Ra mAb and the same pool of MHC I and MHC II-restricted peptides to stimulate memory T cells as well as a second MHC I-restricted pool of peptides (CMV, EBV, Flu) to confirm these results. The survival was evaluated at a somewhat later time-point (day 10) within proliferated and non-proliferated T cells to assess IL7 impact selectivity on restimulated memory T cells. The data show that IL7 significantly increased the proliferation of human memory CD4+ and CD8+ T cells after 10 days of treatment while **it was not the case when IL7-receptor was blocked with the antibody (Figure 7C Left)**. Interestingly, the survival of proliferated cells (Fig 7C middle) was weaker as compared to non-proliferating cells (Fig 7C Right) and **IL7 increased survival only in proliferated cells**. Blocking IL7-receptor with the antibody prevented this beneficial and selective pro-survival effect of IL7 on restimulated human memory T cells. As found in the first part of the experiment, IL7 had no effect again on polyclonal stimulation.

Altogether, these novel experiments demonstrated that IL7/IL7R axis sustains the survival of memory T cells after antigenic rechallenge and that blocking anti-IL7Ra mAb prevent selectively the survival of antigen-specific memory T cells after restimulation. As mentioned in previous question, this point on antigen-specific clonal deletion mechanism of action and its potential consequences in term of relapse (transient if strong re-vaccination) in chronic diseases but also in term of safety profile have been more discussed in the manuscript.

The following paragraph was added in the result section: To decipher how anti- IL-7R α mAb might induce selective antigen-specific memory T cell elimination in-vivo, we first evaluated the impact of IL7 in-vitro on the proliferation and survival of human CD8+ and CD4+ memory T cells after restimulation with MHC I (H1H1 flu) and MHC II (CMV, EBV, Flu, Tetanos) -restricted peptides, respectively, and compared with polyclonal stimulation. While IL7 had no significant impact on the proliferation of human memory CD4 or CD8 T cells after 3 days of culture and only a modest (non-significant) positive effect on memory T cells survival (Figure 7A), at late time-point (after a week of stimulation), IL-7 significantly and strongly improved memory CD4 and CD8 human T cells survival after antigen restimulation (Figure 7B). Blocking IL-7/IL-7R axis using anti-IL-7R α mAb prevented the beneficial effect of IL-7 on memory T cells proliferation and survival (Figure 7C). Interestingly, no significant impact was found on human T cells after polyclonal stimulation while cells proliferated vigorously and displayed very weak survival after 8 or 10 days of stimulation (Figure 7B, 7C). Interestingly, IL7 increased survival only in proliferating human cells (Figure 7C middle) while in the same wells IL7 had no impact on quiescent cells (Figure 7C right). Altogether, the IL7/IL7R axis sustains the survival of memory T cells after antigenic rechallenge and anti-IL7Ra mAbs prevent selectively the survival of antigen-specific memory T cells after restimulation.

The following paragraph was added in the discussion section: Indeed, we found in-vitro that IL-7 controls selectively the survival of proliferating (not quiescent) human memory T cells after antigenic restimulation and not polyclonal stimulation. Altogether, anti-IL-7R α mAb induces selective

elimination of antigen-specific memory T cells after antigen challenge by preventing this anti-apoptotic IL-7/IL-7R axis required by memory T cells to persist after restimulation.

Figure 7

6. Figure 7: As indicated in the figure legend, T cell activation was assessed after ex-vivo culture (48 hours). What is the time (days) indicated in the x axis in panels A and B? I guess these must refer to the time after immunization. In this case, these timepoint indications should be kept the same as in panels C and D (which show time in weeks) to avoid confusion.

The indication of time was harmonized among this Figure now. All time-point indicated are “time after treatment (days) with anti-IL7Ra mAb”

7. While refereeing to the results shown in figure 7B, the authors state “No significant difference was observed after treatment compared to pretreatment activation status” (page 12). The expression of CD69 and CD25 should be shown in cells isolated ex vivo and prior to in vitro treatment (first part of the panels in Figure 7B).

We have generated a Supplementary Figure 10 which shows dotplot expression of CD69 and CD25 by cells isolated (from one representative animals) ex-vivo on unstimulated cells and after in-vitro stimulation. The Figure 7B (now 8B in this revised version) already showed the very low frequency of CD25 and CD69+ cells prior stimulation at day 0, 7 and 14 for all animals.

Supplementary figure 10

8. Page 13: Based on their findings, the authors concluded that antagonist anti-IL7R α mAbs do not provoke general immunosuppression but induce long-term immune tolerance in non-human primates through clonal deletion or intrinsic control of antigen-specific memory T cells after antigen rechallenge. However, the data show that after re-immunization with BCG, there is expansion of antigen-specific cells in animals that were previously treated with site-1/2b antibodies. This finding indicates that antigen-specific cells were not deleted but remained present and capable of responding to the specific antigen after more potent stimulation such as re-vaccination with BCG followed by PPD.

This point has been discussed in previous question. To summarize, the mechanism of action of the anti-IL7R α mAb is to block IL7-mediated survival of memory T cells after antigen restimulation (Figure 7). Chetoui N et al. (Immunology 2010) described previously that in the absence of IL7, human memory T cells enter in apoptosis. This lead *in-vivo* to the selective and long-term disappearance of antigen-specific memory T cells after treatment (Figure 5) without inducing lymphopenia or broad immunosuppression. In parallel, skin inflammation and T-cell infiltration are inhibited along this period (> 1 year). For an unexplained reason, one (over 10) animal did not present inhibition of skin inflammation and, in parallel, this animal did not display either the decrease in IFN γ -secreting cells *ex vivo* after tuberculin restimulation reinforcing the mechanism of action (Supplementary Figure 9). The monthly immunogenic intradermal rechallenges were not able to generate new immune response. However, when we re-vaccinated animals with antigen-bearing live-attenuated bacteria, the immunization might have been strong enough to generate new memory T cells (probably from *de novo* naïve T cells since 1 year after treatment elimination) responding to the antigen *ex-vivo* and hence skin inflammation *in-vivo*.

Reviewers' comments:

Reviewer #1 (Remarks to the Author):

The authors have properly replied to most of my comments. I congratulate them on their efforts.

However, they did not properly address one key question. It may be that IL-7 signaling promotes concomitantly anti- and pro-inflammatory effects (although the pro-inflammatory effects prevail over the putative anti-inflammatory). This is based on the fact, for instance, that despite having pro-inflammatory effects overall, IL-7 stimulation leads to the upregulation of anti-inflammatory genes such as IKZF4 (as shown in supplemental table 2 or suggested in Figure 3B). Consequently, it is possible that one of the mechanisms by which site 1/2b Abs are anti-inflammatory (as opposed to site 1 Abs) is not (only) because of their antagonistic nature (as opposed to the dual antagonistic and agonist effect of site 1 ab) but rather because they nonetheless preserve the ability of IL-7 to upregulate anti-inflammatory genes while site 1 Abs do not. That is, while not acting in an agonistic manner, S1/S2b Abs do not necessarily reverse ALL IL-7-mediated effects and this fact may be not be 'neutral' but rather critical for their anti-inflammatory role.

This possibility (which would not necessarily affect the main message of the manuscript but would reorganize some of the authors' conclusions in a significant way) can easily be tested by a similar experiment to the one the authors described in their rebuttal, but in the presence of IL-7 stimulation. In other words, the question is not whether the Abs alone modulate IKZF4 expression (which the authors now clearly show they do not) but instead whether IL-7-mediated upregulation of IKZF4 can be reversed by the S1 Abs but not by the S1/2b Ab. If differences are observed between S1 and S1/S2b Abs in this regard, the authors should at least discuss this possibility and rephrase the manuscript where appropriate.

Reviewer #2 (Remarks to the Author):

The authors have addressed all my previous questions, have performed additional experiments and have revised their manuscript accordingly. I think that the revised manuscript is appropriate for publication.

Reviewer #1:

The authors have properly replied to most of my comments. I congratulate them on their efforts.

However, they did not properly address one key question. It may be that IL-7 signaling promotes concomitantly anti- and pro-inflammatory effects (although the pro-inflammatory effects prevail over the putative anti-inflammatory). This is based on the fact, for instance, that despite having pro-inflammatory effects overall, IL-7 stimulation leads to the upregulation of anti-inflammatory genes such as IKZF4 (as shown in supplemental table 2 or suggested in Figure 3B). Consequently, it is possible that one of the mechanisms by which site 1/2b Abs are anti-inflammatory (as opposed to site 1 Abs) is not (only) because of their antagonistic nature (as opposed to the dual antagonistic and agonist effect of site 1 ab) but rather because they nonetheless preserve the ability of IL-7 to upregulate anti-inflammatory genes while site 1 Abs do not. That is, while not acting in an agonistic manner, S1/S2b Abs do not necessarily reverse ALL IL-7-mediated effects and this fact may be not be 'neutral' but rather critical for their anti-inflammatory role. This possibility (which would not necessarily affect the main message of the manuscript but would reorganize some of the authors' conclusions in a significant way) can easily be tested by a similar experiment to the one the authors described in their rebuttal, but in the presence of IL-7 stimulation. In other words, the question is not whether the Abs alone modulate IKZF4 expression (which the authors now clearly show they do not) but instead whether IL-7-mediated upregulation of IKZF4 can be reversed by the S1 Abs but not by the S1/2b Ab. If differences are observed between S1 and S1/S2b Abs in this regard, the authors should at least discuss this possibility and rephrase the manuscript where appropriate.

We agree with the comment of the reviewer and performed the experiment requested. We added a new supplementary figure (below) in the manuscript which showed confirmation by RT-qPCR that IKZF4 gene expression is induced by IL-7 and inhibited by site 1 Abs while it is not the case with the site 1/2b Ab. This point has been added in the results section and the discussion was modified accordingly and in view with the hypotheses of the reviewer which indeed turned out to be possible!

Supplementary figure 9: IKZF4 mRNA expression confirmed by RT-qPCR on human PBMC cultured for 3.5 hours with (empty round) or without (full symbols) 5 ng/ml of IL-7 and 10 µg/ml of anti-human IL-7Rα mAbs (blue: site-1/2b IgG4, red: site-1 IgG4#1, green: site-1 IgG1#2). Data were normalized to basal expression in the absence of IL-7 and mAbs. * p<0.05 between indicated groups.

The following sentences have been added in the results section:

All mAbs significantly prevented cluster-1 and cluster-3 modification but without impact on the mixed cluster-2 (Figure 4B, bottom). While site-1/2b mAb was more efficient in-vivo, it sounds to be less efficient in-vitro in preventing some IL-7-induced gene expression modification within the cluster-3, such as for example the anti-inflammatory IKZF4 member of the Ikaros family of transcription factors, implicated in the control of lymphoid development. This result has been confirmed by RT-qPCR (Supplementary Figure 9) and suggests that some anti-inflammatory effect of IL-7 might be conserved by the site-1/2b mAb.

The following sentences has been added in the discussion:

On the other hand, some IL-7-induced anti-inflammatory gene over-expression were efficiently inhibited with site-1 mAbs but not with the site-1/2b mAb (see heatmap in Figure 4B). For example, the Ikaros family member IKZF4 which is highly induced by IL-7 and not clearly affected by site 1/2b Ab, was reported to prevent TH17 polarization (Liu et al JBC 2014), therefore it might be conceivable that some anti-inflammatory actions of IL-7 are differentially inhibited by the two classes of anti-IL7R mAbs and might contribute to the difference observed in-vivo.

Reviewer #2:

The authors have addressed all my previous questions, have performed additional experiments and have revised their manuscript accordingly. I think that the revised manuscript is appropriate for publication.

REVIEWERS' COMMENTS:

Reviewer #1 (Remarks to the Author):

The authors have properly addressed my remaining concern. I congratulate them for the quality of the work.